# VICE: Variational Interpretable Concept Embeddings

**Lukas Muttenthaler**[*]
Machine Learning Group
Technische Universität Berlin
BIFOLD[†]
Berlin, Germany

**Charles Y. Zheng**
Machine Learning Team, FMRI Facility
National Institute of Mental Health
Bethesda, MD, USA

**Patrick McClure**[‡]
Department of Computer Science
Naval Postgraduate School
Monterey, CA, USA

**Robert A. Vandermeulen**
Machine Learning Group
Technische Universität Berlin
BIFOLD[†]
Berlin, Germany

**Martin N. Hebart**
Vision and Computational Cognition Group
MPI for Human Cognitive and Brain Sciences
Leipzig, Germany

**Francisco Pereira**
Machine Learning Team, FMRI Facility
National Institute of Mental Health
Bethesda, MD, USA

## Abstract

A central goal in the cognitive sciences is the development of numerical models for mental representations of object concepts. This paper introduces Variational Interpretable Concept Embeddings (VICE), an approximate Bayesian method for embedding object concepts in a vector space using data collected from humans in a triplet odd-one-out task. VICE uses variational inference to obtain sparse, non-negative representations of object concepts with uncertainty estimates for the embedding values. These estimates are used to automatically select the dimensions that best explain the data. We derive a PAC learning bound for VICE that can be used to estimate generalization performance or determine a sufficient sample size for experimental design. VICE rivals or outperforms its predecessor, SPoSE, at predicting human behavior in the triplet odd-one-out task. Furthermore, VICE's object representations are more reproducible and consistent across random initializations, highlighting the unique advantage of using VICE for deriving interpretable embeddings from human behavior.

## 1 Introduction

Human knowledge about object concepts encompasses many types of information, including function or purpose, visual appearance, encyclopedic facts, or taxonomic characteristics. A central question in cognitive science concerns the representation of this knowledge and its use across different tasks. One approach to this question is inductive and lets subjects list properties for hundreds to thousands of objects [28, 10, 6, 20], yielding large lists of responses about different types of properties. Specifically, objects are represented as vectors of binary properties. While this approach is agnostic to downstream prediction tasks, it may be biased since subjects may leave out important features and/or

---

[*]Also affiliated with the Max Planck Institute for Human Cognitive and Brain Sciences, Leipzig, Germany.
[†]Berlin Institute for the Foundations of Learning and Data, Germany.
[‡]Work was partially done while affiliated with the National Institute of Mental Health, Bethesda, MD, USA.

36th Conference on Neural Information Processing Systems (NeurIPS 2022).

mention unimportant ones. For example, one may forego a general property (e.g., "is an animal") while providing a highly specific fact (e.g., "is found in Florida"). In an alternative, deductive approach, researchers postulate dimensions of interest and subsequently let subjects rate objects in each dimension. Binder et al. [3] employed such an approach and collected ratings for hundreds of objects, verbs, and adjectives. These ratings were gathered over 65 dimensions, reflecting sensory, motor, spatial, temporal, affective, social, and cognitive experiences. Nonetheless, it is desirable to discover object representations that are not biased by the conducted behavioral task and whose dimensions are interpretable without necessitating *a priori* assumptions about their semantic content.

Recently, Zheng et al. [46] and Hebart et al. [19] introduced SPoSE, a model of the mental representations of 1,854 objects in a 49-dimensional space. The model was learned from human judgments about object similarity, where subjects were asked to determine an odd-one-out object in random triplets of objects. SPoSE embedded each object in a vector space so that each dimension is non-negative and sparse (most objects have a close-to-zero entry for a given dimension). The authors showed that the embedding dimensions of objects were interpretable and that subjects could coherently label what the dimensions were "about," ranging from categorical (e.g., animate, food) to functional (e.g., tool), structural (e.g., made of metal or wood), or visual (e.g., coarse pattern). The authors hypothesized that interpretability arose from combining positivity and sparsity constraints so that no object was represented by every dimension, and most dimensions were present for only a few objects. In addition, SPoSE could predict human judgements close to the estimated best attainable performance [46, 19].

Despite its notable performance, SPoSE has several limitations. The first stems from the use of an $\ell_1$ sparsity penalty to promote interpretability. In SPoSE, 6 to 11 dominant dimensions for an object account for most of the prediction performance. These dimensions are different between objects. A potential issue with enforcing SPoSE to have even fewer dimensions is that it may cause excessive shrinkage of the dominant values [1]. Second, when inspecting the distributions of values across objects, most SPoSE dimensions do not reflect the exponential prior induced by the $\ell_1$ penalty. Overcoming this prior may lead to suboptimal performance and inconsistent solutions, specifically in low data regimes. Third, SPoSE uses an ad-hoc criterion for determining the dimensionality of the solution via an arbitrary threshold on the density of each dimension. Finally, SPoSE has no criterion for determining convergence of its optimization process, nor does it provide any formal guarantees on the sample size needed to learn a model of desired complexity.

To overcome these limitations we introduce VICE, a variational inference (VI) method for embedding object concepts with interpretable, sparse, and non-negative dimensions. We start by discussing related work and a description of the triplet task and SPoSE, followed by our presentation of theory and experimental results.

**Contribution 1: VICE solves major limitations of SPoSE** First, VICE encourages shrinkage while allowing for small entry values by using a *spike-and-slab* prior [5, 12, 15, 29, 36, 40, 44]. We deem this more appropriate than an exponential prior, because *importance* – the value an object takes in a dimension – is different from *relevance* – whether the dimension is applicable to that object – and both can be controlled separately. Second, we use VI with a unimodal posterior for representing each object in a dimension which yields a mean value and an uncertainty estimate. While unimodality makes it possible to use the mean values as representative object embeddings, the uncertainty estimates allow us to use a statistical procedure to automatically select the dimensions that best explain the data. Third, we use this procedure to introduce a convergence criterion that reliably identifies *representational stability*, i.e. the consistency in the number of selected dimensions.

**Contribution 2: A PAC bound on the generalization of SPoSE and VICE models** This bound can be used *retrospectively* to provide guarantees about the generalization performance of a converged model. Furthermore, it can be used *prospectively* to determine the sample size required to identify a representation given the number of objects and a maximum possible number of dimensions.

**Contribution 3: Extensive evaluation of model performance across multiple datasets** We compare VICE with SPoSE over three different datasets. One of these datasets contains concrete objects, another consists of adjectives, and the third is composed of food items. Experimentally, we find that VICE rivals or outperforms the performance of SPoSE in modeling human behavior. Moreover, we find that VICE yields dimensions that are much more reproducible, with a lower variance for the number of (selected) dimensions. Both of these measures are particularly important in the cognitive sciences. Lastly, we compare VICE with SPoSE for reduced amounts of data and show that VICE has significantly better performance on all measures.

A `PyTorch` implementation of VICE featuring Continuous Integration is publicly available at `https://github.com/LukasMut/VICE`. The GitHub repository additionally includes code to reproduce all of the experiments that are presented in this paper.

## 2 Related work

Navarro & Griffiths [32] introduced a method for learning semantic concept embeddings from item similarity data. The method infers the number of embedding dimensions using the Indian Buffet Process (IBP) [18]. Their approach relies on continuous-valued similarity ratings rather than discrete forced-choice behavior and is not directly applicable to our setting. It is also challenging to scale the IBP to the number of dimensions and samples in our work [37]. Roads & Love [39] introduced a method for learning object embeddings from behavior in an 8-rank-2 task. Their method predicts behavior from the embeddings by using active sampling [16] to query subjects with the most informative stimuli, yielding an object similarity matrix. The interpretability of embedding dimensions was not considered in this work.

A different approach has been to develop interpretable concept representations from text corpora. Early methods used word embeddings with positivity and sparsity constraints [31]. Later work in this direction used topic model representations of Wikipedia articles about objects [35], transformations of word embeddings into sparse non-negative representations [42, 33], or predictions of properties [10] or dimensions [45]. Others have considered using text corpora in conjunction with imaging data [13, 8]. Finally, Derby et al. [9] introduced a neural network function that maps the sparse feature space of a semantic property norm to the dense space of a word embedding, identifying informative combinations of properties and ranking candidate properties for new words.

## 3 Triplet task

The *triplet task*, also known as the *triplet odd-one-out task*, is used for discovering object concept embeddings from similarity judgments over a set of $m$ different objects. These judgments are collected from human participants who are given queries that consist of a *triplet* of objects (e.g., {"suit", "flamingo", "car"}). Participants are asked to consider the three pairs in a triplet {{"suit","flamingo"}, {"suit","car"}, {"flamingo","car"}}, and to decide which pair is the most similar, leaving the third as the odd-one-out. We assign each object a numerical index, e.g., $1 \leftarrow$ "aardvark", ..., $1854 \leftarrow$ "zucchini". Let $\{y, z\}$ denote the indices in this pair, e.g., $\{y, z\} = \{268, 609\}$ for "suit" and "flamingo." A dataset $\mathcal{D}$ is a set of $n$ ordered pairs of presented triplets and selected pairs. That is, $\mathcal{D} := \left(\{i_s, j_s, k_s\}, \{y_s, z_s\}\right)_{s=1}^{n}$, where $\{y_s, z_s\} \subset \{i_s, j_s, k_s\}$. In Appendix A we discuss in detail why the triplet task appears to be a sensible choice for modeling object similarities in humans.

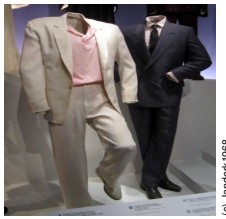 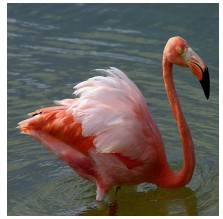 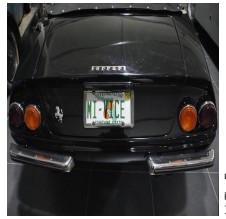

Figure 1: Example triplet containing the objects "suit", "flamingo", and "car" (creative commons images).

## 4 Formal setting

Sparse Positive object Similarity Embedding (SPoSE) [46] is an approach for finding interpretable embedding dimensions from the triplet task. It does so by finding an embedding vector $\mathbf{x}_i = [x_{i1}, \ldots, x_{id}]$ for every object $i$. Let $X$ denote the $m \times d$ matrix $(\mathbf{x}_1, \ldots, \mathbf{x}_m)$ and $S := XX^T$ be the similarity matrix, where $S_{ij}$ denote its entry at $i, j$. The probability of choosing $\{y_s, z_s\}$ as the most

similar pair of objects, given an object triplet $\{i_s, j_s, k_s\}$ and the embedding matrix $X$, is modeled as

$$p(\{y_s, z_s\}|\{i_s, j_s, k_s\}, X) := \frac{\exp(S_{y_s, z_s})}{\exp(S_{i_s, j_s}) + \exp(S_{i_s, k_s}) + \exp(S_{j_s, k_s})}. \quad (1)$$

At this point we remark that each triplet $\{i_s, j_s, k_s\}$ is chosen uniformly at random from the collection of all possible sets of object triplets $\mathcal{T}$. That is, $\{i_1, j_1, k_1\}, \ldots, \{i_n, j_n, k_n\} \overset{\text{i.i.d.}}{\sim} \mathcal{U}(\mathcal{T})$. This is precisely stated in Appendix B.1. SPoSE uses maximum a posteriori (MAP) estimation with a Laplace prior under a non-negativity constraint (equivalent to an exponential prior) to find the most likely embedding $X$ for the training data $\mathcal{D}$. This leads to the training objective

$$\arg\min_{X \geq 0} \ -\log p(\mathcal{D}|X) + \lambda \sum_{i=1}^{m} \sum_{j=1}^{d} |X_{ij}|,$$

where $\log p(\mathcal{D}|X) = \text{constant} + \sum_{s=1}^{n} \log p(\{y_s, z_s\}|\{i_s, j_s, k_s\}, X)$ (see Appendix B.1 for a full derivation of the log-likelihood function) and $\lambda$ is determined using cross-validation.

# 5 VICE

In contrast to SPoSE, we use mean-field VI [4] instead of a MAP estimate for approximating the posterior probability $p(X|\mathcal{D})$. In VICE we impose additional constraints on the embedding matrix $X$ by using a prior that encourages shrinkage while allowing entries in $X$ to be close to zero.

---

**Algorithm 1** VICE optimization for individual triplets for a single training epoch

---

**Input:** $\mathcal{D}, \theta, \alpha$        $\triangleright$ Recall that $\mathcal{D} := (\{i_s, j_s, k_s\}, \{y_s, z_s\})_{s=1}^{n}$, and $\alpha$ is a learning rate
    $\theta := \{\mu, \sigma\}$                                             $\triangleright \mu, \sigma \in \mathbb{R}^{m \times d}$
    **for** $s \in \{1, \ldots, n\}$ **do**
       $\epsilon \sim \mathcal{N}(0, I)$         $\triangleright$ Draw i.i.d Gaussian noise where $\epsilon \in \mathbb{R}^{m \times d}$ and $\epsilon_{ij} \overset{\text{i.i.d.}}{\sim} \mathcal{N}(0, 1)$
       $X_{\theta, \epsilon} := \mu + \sigma \odot \epsilon$         $\triangleright$ Apply reparameterization trick; $\odot$ is the Hadamard product
       $S := [X_{\theta, \epsilon}]_{+}[X_{\theta, \epsilon}]_{+}^{T}$         $\triangleright$ Compute similarity matrix using the non-negative parts of $X_{\theta, \epsilon}$
       $\mathcal{L}_{\text{data}}(X_{\theta, \epsilon}) \triangleq \log \left[ \frac{\exp(S_{y_s, z_s})}{\exp(S_{i_s, j_s}) + \exp(S_{i_s, k_s}) + \exp(S_{j_s, k_s})} \right]$     $\triangleright$ Data log-likelihood term
       $\mathcal{L}_{\text{complexity}}(X_{\theta, \epsilon}) \triangleq \frac{1}{n} \left[ \log q_\theta(X_{\theta, \epsilon}) - \log p(X_{\theta, \epsilon}) \right]$    $\triangleright$ KL divergence/Regularization term
       $\mathcal{L}_{\text{total}}(X_{\theta, \epsilon}) \triangleq \mathcal{L}_{\text{complexity}}(X_{\theta, \epsilon}) - \mathcal{L}_{\text{data}}(X_{\theta, \epsilon})$
       $\mu \leftarrow \mu - \alpha \nabla_\mu \mathcal{L}_{\text{total}}(X_{\theta, \epsilon})$                     $\triangleright$ Update the embedding means
       $\sigma \leftarrow \sigma - \alpha \nabla_\sigma \mathcal{L}_{\text{total}}(X_{\theta, \epsilon})$            $\triangleright$ Update the embedding standard deviations
    **end for**
**Output:** $\theta$                              $\triangleright$ Return the optimized set of model parameters $\theta$

---

## 5.1 Variational Inference

For VICE we consider approximating $p(X|\mathcal{D})$ with a variational distribution, $q_\theta(X)$, where $q_\theta \in \mathcal{Q}$, and $\theta$ is optimized to minimize the KL divergence to the true posterior, $p(X|\mathcal{D})$. In VICE the KL divergence objective function (derived in Appendix B.2) is

$$\arg\min_{\theta} \ \mathbb{E}_{q_\theta(X)} \left[ \frac{1}{n} \left( \log q_\theta(X) - \log p(X) \right) - \frac{1}{n} \sum_{s=1}^{n} \log p \left( \{y_s, z_s\}|\{i_s, j_s, k_s\}, X \right) \right]. \quad (2)$$

**Variational distribution** VI requires a choice of a parametric variational distribution $q \in \mathcal{Q}$. For VICE we use a Gaussian distribution with a diagonal covariance matrix $q_\theta(X) = \mathcal{N}(\mu, \text{diag}(\sigma^2))$ where $\theta = \{\mu, \sigma\}$. Therefore, each embedding dimension has a *mean* and a *standard deviation*. We deem a Gaussian variational distribution appropriate for a variety of reasons. First, under certain conditions, the posterior is Gaussian in the infinite-data limit [24]. Second, a unimodal variational distribution makes it possible to use $\mu_1, \ldots, \mu_m$ as representative object embeddings. A fixed representation is useful for downstream use cases that rely on a single embedding vector for each object. Use cases range from using embeddings as targets in a regression task over unsupervised

clustering of embeddings to interpreting embeddings. More complex variational families are clearly not as practical for this. Third, a Gaussian posterior is a computationally convenient choice.

Similarly to Titsias & Lázaro-Gredilla [43], we use a Monte Carlo approximation of Equation 2 by sampling a limited number of $X$s from $q_\theta(X)$ during training. We generate $X$ with the reparameterization trick [23, 38], $X_{\theta,\epsilon} = \mu + \sigma \odot \epsilon$, where $\epsilon \in \mathbb{R}^{m \times d}$ is entrywise $\mathcal{N}(0,1)$, and $\odot$ denotes the Hadamard (element-wise) product. This leads to the objective

$$\arg\min_\theta \; \frac{1}{nR} \sum_{r=1}^R \left( \log q_\theta(X_{\theta,\epsilon^{(r)}}) - \log p(X_{\theta,\epsilon^{(r)}}) - \sum_{s=1}^n \log p(\{y_s, z_s\} | \{i_s, j_s, k_s\}, [X_{\theta,\epsilon^{(r)}}]_+) \right). \tag{3}$$

We apply a ReLU function, denoted by $[\cdot]_+$, to the sampled $X_{\theta,\epsilon}$ values to guarantee that $X_{\theta,\epsilon} \in \mathbb{R}_+^{m \times d}$. As commonly done in the Dropout and Bayesian Neural Network literature [41, 5, 14, 27], we set $R = 1$ for computational efficiency during the optimization process. The optimization is outlined for individual triplets (i.e., where $B = 1$) for a single training epoch in Algorithm 1.

**Posterior probability estimation** Computational efficiency at inference time is not as critical as it is during training. Therefore, we can get a better estimate of the posterior probability distribution over the three possible odd one-one-out choices by letting $R \gg 1$. Using the optimized variational posterior, $q_{\hat\theta(X)}$, we approximate the probability distribution with a Monte Carlo estimate [17, 5, 27, 4] from $R$ samples $X^{(r)} = X_{\hat\theta,\epsilon^{(r)}}$ for $r = 1, \ldots, R$, yielding

$$\hat p(\{y, z\} | \{i, j, k\}) := \frac{1}{R} \sum_{r=1}^R p(\{y, z\} | \{i, j, k\}, X^{(r)}). \tag{4}$$

**Spike-and-slab prior** As discussed above, SPoSE induces sparsity through an $\ell_1$ penalty which, along with the non-negativity constraint, is equivalent to using an exponential prior. Through examination of the publicly available histograms of weight values in the two most important SPoSE dimensions (see Figure 2 in Appendix C.3), we observed that the dimensions did not resemble an exponential distribution. Instead, they contained a *spike* of probability at zero and a wide *slab* of probability for the non-zero values. To model this, we use a spike-and-slab Gaussian mixture prior [5, 12, 15, 21, 26],

$$p(X) = \prod_{i=1}^m \prod_{j=1}^d (\pi_{\text{spike}} \mathcal{N}(X_{ij}; 0, \sigma_{\text{spike}}^2) + (1 - \pi_{\text{spike}}) \mathcal{N}(X_{ij}; 0, \sigma_{\text{slab}}^2)), \tag{5}$$

which encourages shrinkage. This prior has three parameters, $\sigma_{\text{spike}}$, $\sigma_{\text{slab}}$, and $\pi_{\text{spike}}$. $\pi_{\text{spike}}$ is the probability that an embedding dimension is drawn from the *spike* Gaussian. Since spike and slab distributions are mathematically interchangeable, by convention we require that $\sigma_{\text{spike}} \ll \sigma_{\text{slab}}$.

## 5.2 Dimensionality reduction and convergence

For interpretability purposes it is desirable for the object embedding dimensionality, $d$, to be small. In contrast to SPoSE, which employs a user-defined threshold to prune dimensions, VICE exploits the uncertainty estimates for embedding values to select a subset of informative dimensions.

The pruning procedure works by assigning importance scores to each of the $d$ dimensions, which reflect the number of objects that we can confidently say have a non-zero weight in a dimension. To compute the score, we use the variational embedding for each object $i$ and dimension $j$ – location $\mu_{ij}$ and scale $\sigma_{ij}$ parameters – to compute the posterior probability that the weight is truncated to zero according to the left tail of a Gaussian distribution with that location and scale (see §5.1). This gives us a posterior probability of the weight being zero for each object within a dimension [17]. To calculate the overall importance of a dimension, we estimate the number of objects that have non-zero weights, while controlling the False Discovery Rate [2] with $\alpha = 0.05$. We define the importance of each dimension $j$ to be the number of objects for which $P(X_{ij} > 0) \geq .95$ holds. After convergence, we prune the model by removing dimensions with 5 or fewer statistically significant objects. This is a commonly used reliability threshold in semantic property norms (e.g., [28, 10]).

In gradient-based optimization, the gradient of an objective function with respect to the parameters of a model, $\nabla \mathcal{L}(\theta)$, is used to iteratively find parameters $\hat\theta$ that minimize that function. We use a *representational stability* criterion to determine convergence. That is, the optimization process

halts when the number of identified dimensions - as described above - has not changed by a *single* dimension over the past $L$ epochs (e.g., $L = 500$). Given that our goal is to find stable estimates of the number of dimensions, we considered this to be more appropriate than other convergence criteria such as evaluating the cross-entropy error on a validation set or evidence-based criteria [25, 11]. For further details on convergence and the optimization process see Appendix C.1.

## 6 Sample complexity bound

We use statistical learning theory to obtain estimates of the sample size needed to appropriately constrain VICE (and SPoSE) models. These estimates can be used *retrospectively* to obtain probabilistic guarantees on the generalization of a trained model to unseen data, or *prospectively* to decide how much data to collect for a study. The bound presented here applies to any embedding matrix, $X$, using the model in Equation 1. When using this bound to analyze VICE, we extract $\mu$ and use it as a fixed embedding to predict the most likely odd-one-out for a query triplet, rather than sampling from the variational distribution.

To obtain a useful bound we make two assumptions. First, we assume that there exists an upper bound $M$ on the largest value in the embedding. Second, we assume that the embeddings obtained by either SPoSE or VICE can be quantized coarsely with marginal losses in predictive performance. We can choose a discretization scale $\Delta$, e.g., $\Delta = 0.5$, and round embedding values to a non-negative integer multiple of $\Delta$. While we employ this quantization primarily to use learning theory bounds for finite hypothesis classes, it could have benefits for interpretation as well (e.g., a dimension could consist of labels such as zero (0), very low (0.5), low (1), medium (1.5), high (2)). In all datasets we have used, SPoSE and VICE embeddings with reasonable priors had values below 2.7 (see Appendix F.2). Given an upper bound $M$ and a discretization scale $\Delta$, all entries of $X$ are limited to the set $\mathbb{A} := \{0, \Delta, \ldots, (k-1)\Delta, k\Delta\}$. The following bound tells us that, given enough samples, the true error rate is not much worse than the estimated error rate for such discretized $X$ matrices, with high probability.

**Proposition 6.1.** *Given $\delta > 0$, $\epsilon > 0$, and $n \geq \left(md\log(k+1) + \log(1/\delta)\right)/\left(2\epsilon^2\right)$ training samples $(\{y_s, z_s\}, \{i_s, j_s, k_s\})_{s=1}^{n}$, then*

$$P\left(\sup_{X \in \mathbb{A}^{m \times d}} \hat{R}(X) - R(X) < \epsilon\right) \geq 1 - \delta,$$

*where*

$$\hat{R}(X) := \frac{1}{n}\sum_{s=1}^{n} \mathbb{1}\left(\{y_s, z_s\} \neq \arg\max_{\{y,z\}} p\left(\{y, z\}|\{i_s, j_s, k_s\}, X\right)\right)$$

*and*

$$R(X) := P_{(\{y', z'\}, \{i', j', k'\})}\left(\{y', z'\} \neq \arg\max_{\{y,z\}} p\left(\{y, z\}|\{i', j', k'\}, X\right)\right).$$

*Proof sketch.* This bound follows from applying Hoeffding's inequality to the $(k+1)^{m \times d}$ elements of $\mathbb{A}^{m \times d}$ combined with a union bound. This proof is virtually identical to that of Theorem 2.13 in Mohri et al. [30]. A complete proof can be found in Appendix D. $\qquad\square$

To use the bound we first decide on the number of quantization steps $k = \lceil M/\Delta \rceil$. The probability of violating the bound, $\delta$, is analogous to the Type I error control in hypothesis testing and is often set to 0.05. Since $n$ in our proposition depends very weakly on $\delta$, it is convenient to use $\delta = 0.001$. The error tolerance $\epsilon$ *does* have a major effect on the sample size estimate provided by the bound. For the bound to be practically useful, $\epsilon$ has to be smaller than the difference between the training error – average zero-one loss over the training set examples – and random guessing error. For the datasets we use in this paper we observed a difference of $0.2 - 0.3$. A conservative choice would halve the accuracy gap of 0.2, giving $\epsilon = 0.1$. Together, with $k = 4$ (from above), these values result in a prospective sample size of $n \geq 50 \cdot (md\log(5) + \log(1000))$. To use the bound retrospectively, we fix $n$, $\delta$ while varying $\Delta$ (and consequently $k$) to get a guarantee on $\epsilon$, which in turn yields a probabilistic upper bound on the generalization error for an embedding, $R(X) \leq \hat{R}(X) + \epsilon$. For more details, see Algorithm 1 in Appendix F.1.

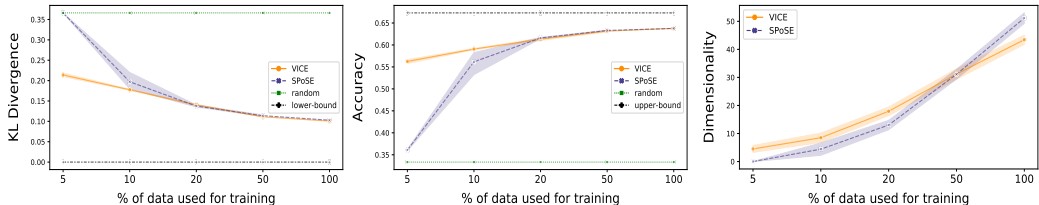

Figure 2: VICE vs. SPoSE. **Left**: KL divergences between model output and human probability distributions. **Center**: Triplet task prediction accuracies. **Right**: Number of identified embedding dimensions. VICE and SPoSE were each trained on differently sized subsets of the THINGS training data. Error bands visualize 95% CIs across the different random initializations and partitions of the data.

## 7 Experiments

**Data** We performed experiments for three triplet datasets: THINGS (used to develop SPoSE [46, 19]), ADJECTIVES[4], and FOOD [7] [5]. THINGS and ADJECTIVES contain random samples from all possible triplet combinations for 1,854 objects and 2,372 adjectives, respectively. THINGS and ADJECTIVES are each comprised of a large training dataset which contains no "repeats," i.e., there is only one human response for each triplet ($\sim 1.5$M triplets for THINGS and $\sim 800$K triplets for ADJECTIVES). For each of these training datasets, 10% of the triplets are assigned to a predefined validation set. The test sets for both contain the results for 1,000 random triplets that are not contained in the training data, with 25 repeats for each triplet. FOOD contains data for every possible triplet combination for 36 objects and repeats for some triplets. We partitioned this dataset into train (45%), validation (5%), and test (50%) sets with disjoint triplets. Appendix G provides details about stimuli, data collection, and quality control.

**Experimental setup** We implemented both SPoSE and VICE in `PyTorch` [34] using Adam [22] with $\eta = 0.001$. To find the best VICE hyperparameter combination, we performed a grid search over $\sigma_{\text{spike}}, \sigma_{\text{slab}}, \pi_{\text{spike}}$, optimizing Equation 3 and evaluating each model on the validation set. All experiments were performed over 20 different random initializations. For VICE we observed that the hyperparameter values with the lowest average cross-entropy error on the validation set were highly similar across the different datasets (see Table 1 in Appendix E). For more information about the experimental setup, including details about training, weight initialization, hyperparameter grid, and optimal combination, see Appendix E.

**Evaluation measures** VICE estimates $p(\{y, z\}|\{i, j, k\})$, the model's softmax probability distribution over a given triplet (see Equation 4). The first evaluation measure we consider is the prediction *accuracy* with respect to the correct human choice $\{y, z\}$, where the predicted pair is $\arg\max_{\{y,z\}} \hat{p}(\{y, z\}|\{i, j, k\})$. We estimate an upper bound on the prediction accuracy by using the repeats in the test set (see Appendix E). A Bayes-optimal classifier would predict the majority outcome for any triplet, whereas random guessing gives an accuracy of $1/3$. Note that the triplet task is subjective and thus there is not necessarily a definitive correct answer for a given triplet. If a test set contains multiple responses for a triplet, it provides information about the relative similarities of the three object pairs. This lets us calculate probability distributions over answers for each triplet. Approximating these distributions closely is important in cognitive science applications. Therefore, we additionally evaluate the models by computing the KL divergences between the models' posterior probability estimates $\hat{p}(\{y, z\}|\{i, j, k\})$ (see Equation 4) and the human probability distributions inferred from the test set across all test triplets and report the mean.

### 7.1 Results on THINGS

**Full dataset** For this experiment we compared a representative model from VICE and SPoSE. The representative models were chosen to be those with the median cross-entropy error on the validation set over the 20 seeds. For VICE, we set the number of Monte Carlo samples to $R = 50$. On the test set, VICE and SPoSE achieved similar prediction accuracies of $0.638$ and $0.637$ respectively (estimated upper bound for accuracy was $0.673$). Likewise, VICE and SPoSE achieved similar

---

[4]ADJECTIVES is not yet published, but was shared with us by Shruti Japee.
[5]FOOD was shared with us by Jason Avery.

average KL divergences of 0.100 and 0.103 respectively (random guessing KL divergence was 0.366). The differences between the median model accuracies and KL divergences were not statistically significant according to a two-sided paired $t$-test over individual test triplets. Hence, VICE and SPoSE predicted triplets approximately equally well when trained on the full dataset. This is not surprising since Bayesian methods based on Monte Carlo sampling become more like deterministic maximum likelihood estimation in the infinite data limit. If $n \to \infty$, then the left-hand side in the expectation of Equation 2, which accounts for the contribution of the prior, goes to zero. Conversely, the effects of the prior are more prominent when $n$ is small, as we show in the following section.

**Data efficiency experiments** Performance on small datasets is important in cognitive science, since behavioral experiments often have small sample sizes (e.g., tens to hundreds of volunteer in-lab subjects) compared to THINGS, which is particularly large, or they can be costly to collect. To test whether VICE can model the data better than SPoSE in small sample regimes, we performed experiments training on smaller subsets with sizes equal to 5%, 10%, 20%, and 50% of the training dataset. For each size, we divided the training set into equal partitions and performed an experiment for each partition, thereby giving multiple results for each subset size. Validation and held-out test sets remained unchanged. In Figure 2 we show the average KL divergence (left) and prediction accuracy (middle) across data partitions and 20 random seeds (used to initialize model parameters) for various dataset sizes. The average performance across partitions was used to compute the confidence intervals (CIs). The differences in prediction accuracies and KL divergences between VICE and SPoSE were notable for the 5% and 10% data subsets. In the former, SPoSE predicted only slightly better than random guessing; in the latter, SPoSE showed a large variability between random seeds and data partitions, as can be seen in the 95% CIs. In comparison, VICE showed small variation in the two performance metrics across random seeds and performed much better than random guessing. The differences between VICE and SPoSE for the 5% and 10% subset scenarios were statistically significant according to a two-sided paired $t$-test comparing individual triplet predictions between median models ($p < 0.001$). For the full training dataset, VICE used significantly fewer dimensions ($p < 0.001$; unpaired sign test) than SPoSE to achieve comparable performance.

## 7.2 Other results

**Other datasets** On the ADJECTIVES test set, VICE and SPoSE had accuracies of 0.559 and 0.562, respectively (estimated upper bound was 0.607), and KL divergences of 0.083 and 0.088, respectively (random guessing was 0.366). On the FOOD test set, VICE and SPoSE median models had accuracies of 0.693 and 0.698, respectively. The differences in accuracy (and KL divergence for ADJECTIVES) between VICE and SPoSE for both datasets were not statistically significant according to two-sided paired $t$-tests on the median model predictions across triplets on the test set. However, for FOOD, VICE used significantly fewer dimensions than SPoSE to achieve similar performance (see Table 1).

**Hyperparameters** We observed that VICE's performance is fairly insensitive to hyperparameter selection. Using default hyperparameters, $\sigma_{\text{spike}} = 0.25$, $\sigma_{\text{slab}} = 1.0$, $\pi_{\text{sigma}} = 0.5$, yields an accuracy score within 0.015 of the best cross-validated model in the full dataset setting, on all three datasets.

## 7.3 Reproducibility and Representational stability

**Reproducibility** As mentioned in the introduction, a key criterion for learning concept representations beyond predictive performance is *reproducibility*, i.e., learning similar representations for different random initializations on the same training data. To evaluate this, we compared the learned embeddings from 20 differently initialized VICE and SPoSE models. The first aspect of reproducibility we investigate is whether the models yield a consistent number of embedding dimensions across random seeds.

As reported in Table 1, VICE yielded fewer dimensions than SPoSE with less variance in the number of dimensions for all three datasets. The embedding dimensionality was significantly smaller according to an independent $t$-test for THINGS ($p < 0.001$) and FOOD ($p < 0.001$), but not for ADJECTIVES ($p = 0.108$). The difference in the standard deviations for ADJECTIVES was statistically significant according to a two-sided $F$-test ($p = 0.002$), but was not statistically significant for THINGS ($p = 0.283$) or FOOD ($p = 0.378$). In addition, we observed that the identified dimensionality is consistent as long as $d$ is chosen to be sufficiently large (see Appendix E).

Table 1: Reproducibility of VICE and SPoSE. Reported are the means and standard deviations with respect to selected dimensions and the average reproducibility score of dimensions (in %) across random seeds. Bold means VICE performed statistically significantly better with $\alpha = 0.05$.

| DATA\ METRIC | VICE | | SPoSE | |
| --- | --- | --- | --- | --- |
| | Selected Dims. | Reproducibility | Selected Dims. | Reproducibility |
| THINGS | **44** (1.59) | **87.01**% | 52 (1.82) | 81.30% |
| ADJECTIVES | 21 (**0.77**) | 76.76% | 22 (1.53) | 71.64% |
| FOOD | **5** (0.95) | **87.38**% | 16 (1.02) | 62.88% |

The second aspect to reproducibility we examine is the degree to which the identified dimensions are similar across random initializations up to a permutation. To calculate the number of reproducible dimensions, we associated each embedding dimension of a model with the most similar embedding dimension across the other models. We quantify reproducibility of a dimension as the average Pearson correlation between one dimension and its best match across the 19 remaining models. In Table 1 we report the average relative number of dimensions with a correlation $> 0.8$ across models. The embedding dimensions in VICE were more reproducible than those in SPoSE. The difference in average reproducibility scores was statistically significant according to a two-sided, independent $t$-test for FOOD ($p = 0.030$) and THINGS ($p = 0.008$), but not for ADJECTIVES ($p = 0.466$).

**Representational stability** For all three datasets, VICE found reproducible and stable dimensions across all 20 random initializations and converged when the embedding dimensionality had not changed over the past $L = 500$ epochs. The median number of epochs to achieve representational stability was comparable for the similarly-sized THINGS and ADJECTIVES datasets, but occurred later for FOOD (likely due to the smaller number of gradient updates per epoch). In Appendix C.1 we show plots that demonstrate that the convergence criterion as defined in §5.2 worked reliably for all datasets.

## 7.4 Interpretability

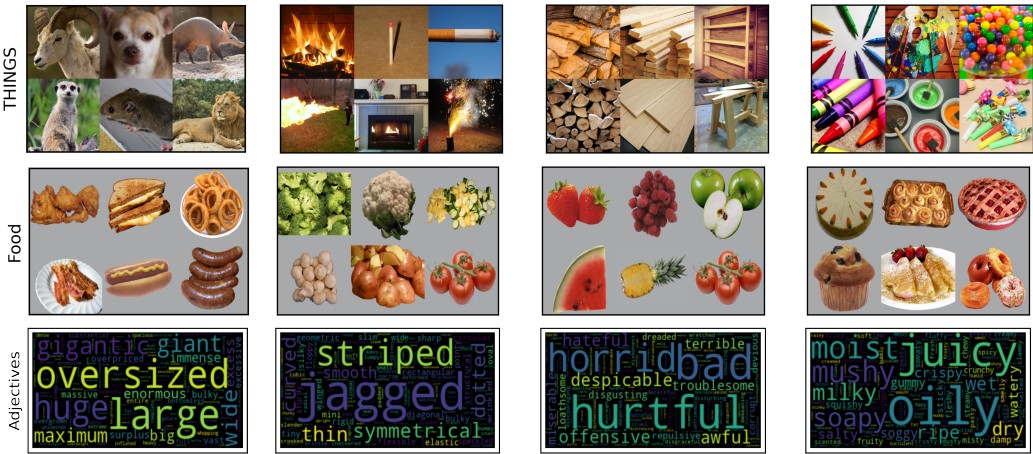

Figure 3: Four example VICE dimensions showing the top six objects for THINGS (*top*) and FOOD (*middle*), and wordclouds for ADJECTIVES (*bottom*).

One of the benefits of SPoSE is the interpretability of the dimensions of its concept embeddings which was evidenced through experiments with humans [19]. VICE dimensions are equally interpretable due to similar constraints on its embedding space. Therefore, it is easy to sort objects within a dimension of the VICE mean embedding matrix $\mu$ in descending order, to obtain human judgments of what an embedding dimension represents. To illustrate the interpretability of VICE we show in Figure 3 four example dimensions of a representative model for THINGS, FOOD, and ADJECTIVES. For THINGS the dimensions appear to represent *categorical*, *functional*, *structural*, and *color-related* information. For FOOD the dimensions appear to be a combination of *fried*, *vegetable*, *fruit*, and *sweet* food items. For ADJECTIVES the dimensions seem to reflect *size/magnitude*, *visual appearance*,

*negative valence*, and *sensory* adjectives. In Appendix H we show VICE dimensions for THINGS only, since manuscripts about the other datasets are still in preparation, and, hence, cannot be shown publicly.

## 8    Limitations

The goal of our method is to identify general mental representations of objects. The dimensions identified by a model reflect semantic characteristics that explain task performance for many subjects in the experimental subject population (here, Amazon Mechanical Turk subjects in the United States). As such, it is possible that they reflect biases widespread in that population. Furthermore, the choice of population may affect the identified dimensions. That is, a chéf may classify food items differently from a lay subject, and a linguist would likely have a more complex representation of an adjective. The effects of expertise or developmental stage in mental representations are of obvious interest to cognitive scientists. Therefore, we envision further research in those areas, which may additionally provide some indication of how widely representations can vary.

## 9    Conclusion

One of the central goals in the cognitive sciences is the development of computational models of mental representations of object concepts. Such models may serve as a component for other behavioral prediction models or as a basis for identifying the nature of concept representations in the human brain. In this paper we introduced VICE, a novel VI approach for learning interpretable object concept embeddings by modeling human behavior in an triplet odd-one-out task. We showed that VICE predicts human behavior close to the estimated best attainable performance across three datasets and that VICE outperforms a competing method, SPoSE, in low data regimes. In addition, VICE has several characteristics that are desirable for scientific use. It has an automated procedure for determining the number of dimensions sufficient to explain the data, which further enables the detection of convergence during training. This leads to better model reproducibility across different random initializations and hyperparameter settings. As a result, VICE can be used out-of-the-box without requiring to perform an extensive search over random seeds or tuning model hyperparameters. Finally, we introduced a PAC learning bound on the generalization performance for a VICE model. Although VICE assumes a shared mental representation across participants - akin to much of the literature -, we believe that the VI framework can be leveraged to model inter-individual differences, which we plan to do in future work.

**Acknowledgments**

LM and RV acknowledge support by the Federal Ministry of Education and Research (BMBF) for the Berlin Institute for the Foundations of Learning and Data (BIFOLD) (01IS18037A). LM and MNH acknowledge support by a Max Planck Research Group grant awarded to MNH by the Max Planck Society. FP, CZ, and PM acknowledge the support of the National Institute of Mental Health Intramural Research Program (ZIC-MH002968). PM acknowledges the support of the Naval Postgraduate School's Research Initiation Program. This study utilized the high-performance computational capabilities of the Biowulf Linux cluster at the National Institutes of Health, Bethesda, MD (http://biowulf.nih.gov) and the Raven and Cobra Linux clusters at the Max Planck Computing & Data Facility, Garching, Germany (https://www.mpcdf.mpg.de/services/supercomputing/). The authors would like to thank Chris Baker for useful initial discussions, Shruti Japee, Jason Avery, and Alex Martin for sharing data, and Erik Daxberger, Lorenz Linhardt, Adrian Hill, Niklas Schmitz, and Marco Morik for valuable feedback on earlier versions of the paper.

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
