# Supplementary for "VICE: Variational Interpretable Concept Embeddings"

**Lukas Muttenthaler**[*]
Machine Learning Group
Technische Universität Berlin
BIFOLD[†]
Berlin, Germany

**Charles Y. Zheng**
Machine Learning Team, FMRI Facility
National Institute of Mental Health
Bethesda, MD, USA

**Patrick McClure**[‡]
Department of Computer Science
Naval Postgraduate School
Monterey, CA, USA

**Robert A. Vandermeulen**
Machine Learning Group
Technische Universität Berlin
BIFOLD[†]
Berlin, Germany

**Martin N. Hebart**
Vision and Computational Cognition Group
MPI for Human Cognitive and Brain Sciences
Leipzig, Germany

**Francisco Pereira**
Machine Learning Team, FMRI Facility
National Institute of Mental Health
Bethesda, MD, USA

## A  Triplet task

The triplet task compares favorably to other possible alternatives for eliciting human pairwise object similarity judgments. Asking for similarity ratings directly, on a numeric or qualitative scale, introduces the difficulty of differing inter-individual calibration of ratings. Having a choice task reduces this inter-individual variability by reducing the number of possible actions the user can make, e.g., three in the triplet task. Within choice tasks, there are also $k$-way forced choice tasks which specify a reference object and query the user for the most similar object to the reference among a list of $k$ candidate objects. It seems likely that the choice in the forced choice task is not strictly driven by pairwise item similarities between the choices and the reference, but that the reference may influence the selection of features used to assess item similarities. For instance, the subject may choose the most prominent feature associated with the reference, and then make a uni-dimensional assessment of which candidate scores highest on that feature. This is less likely to be an issue in the triplet task because with no reference object, the subject has to consider all three pairs of items to evaluate which pair has greater similarity than the other two pairs. In particular, it would be less feasible to try to find a single feature that would be relevant for all three pairs of objects. Hence, we speculate that the triplet task may promote more multidimensional similarity assessments than the $k$-way forced-choice task.

Many studies find it convenient to pool triplet tasks from multiple participants. This could be justified under an assumption where subjects within a population all share the same common mental representations of items, which in turn determine the pairwise similarities that govern the choice of the odd-one-out selection. Alternatively, it could be the case that the representation of each item has random variation, both within a subject and across subjects. In such a scenario, it could be useful

---

[*]Also affiliated with the Max Planck Institute for Human Cognitive and Brain Sciences, Leipzig, Germany.
[†]Berlin Institute for the Foundations of Learning and Data, Germany.
[‡]Work was partially done while affiliated with the National Institute of Mental Health, Bethesda, MD, USA.

36th Conference on Neural Information Processing Systems (NeurIPS 2022).

to use a model which assumes that a *probabilistic* (rather than a *deterministic*) embedding governs the triplet choices. As we discussed theoretically in §5 and showed empirically in §7, variational bayesian inference is one way that appears appropriate to model such a probabilistic embedding.

# B   Objective function

## B.1   Probability model and the log-likelihood function

In the following we define the log-likelihood function of the data given the embedding matrix, $\log p(\mathcal{D}|X)$. Recall that $\mathcal{D} := \left(\{i_s, j_s, k_s\}, \{y_s, z_s\}\right)_{s=1}^{n}$. For simplicity, we assume that each triplet $\{i_s, j_s, k_s\}$ is chosen uniformly at random from the collection of all possible sets of object triplets $\mathcal{T}$. That is, $\{i_1, j_1, k_1\}, \ldots, \{i_n, j_n, k_n\} \overset{\text{i.i.d.}}{\sim} \mathcal{U}(\mathcal{T})$. Hence, the probability for choosing a triplet, $p(\{i_s, j_s, k_s\})$, is the same for all $\mathcal{D}_s \in \{\mathcal{D}_1, \ldots, \mathcal{D}_n\}$ and can therefore be treated as a coefficient for computing $p(\mathcal{D}|X)$.

For our probability model the log-likelihood of the data given the embedding matrix, $\log p(\mathcal{D}|X)$, can then be defined as

$$
\begin{aligned}
\log p(\mathcal{D}|X) &= \log \prod_{s=1}^{n} p(\mathcal{D}_s|X) \\
&= \log \prod_{s=1}^{n} p(\{y_s, z_s\}, \{i_s, j_s, k_s\}|X) \\
&= \log \prod_{s=1}^{n} p(\{y_s, z_s\}|\{i_s, j_s, k_s\}, X) p(\{i_s, j_s, k_s\}|X) \quad\quad (1) \\
&= \log \prod_{s=1}^{n} p(\{y_s, z_s\}|\{i_s, j_s, k_s\}, X) p(\{i_s, j_s, k_s\}).
\end{aligned}
$$

Recall that $p(\{i_s, j_s, k_s\})$ is a coefficient that is the same for all $s \in \{1, \ldots, n\}$. We let $C := p(\{i_s, j_s, k_s\})$ and thus we can further rewrite,

$$
\begin{aligned}
\log \prod_{s=1}^{n} p(\{y_s, z_s\}|\{i_s, j_s, k_s\}, X) p(\{i_s, j_s, k_s\}) &= \log \left[ C^n \prod_{s=1}^{n} p(\{y_s, z_s\}|\{i_s, j_s, k_s\}, X) \right] \\
&= \log C^n + \log \prod_{s=1}^{n} p(\{y_s, z_s\}|\{i_s, j_s, k_s\}, X) \\
&= n \log C + \sum_{s=1}^{n} \log p(\{y_s, z_s\}|\{i_s, j_s, k_s\}, X),
\end{aligned}
$$

where (6) follows from the chain rule of probability. Note that the constant $n \log C$ can be ignored in the minimization of the VICE objective function, whose derivation is outlined in Appendix B.2.

## B.2   KL divergence

In VI, one minimizes the KL divergence between $q_\theta(X)$, the variational posterior, and, $p(X|\mathcal{D})$, the true posterior,

$$
\arg \min_{\theta} D_{\text{KL}}(q_\theta(X) \| p(X|\mathcal{D})),
$$

where

$$D_{\mathrm{KL}}(q_\theta(X)\|p(X|\mathcal{D})) = \mathbb{E}_{q_\theta(X)}\left[\log\frac{q_\theta(X)}{p(X|\mathcal{D})}\right]$$

$$= \mathbb{E}_{q_\theta(X)}\left[\log q_\theta(X) - \log p(X|\mathcal{D})\right]$$

$$= \mathbb{E}_{q_\theta(X)}\left[\log q_\theta(X) - \log\frac{p(\mathcal{D}|X)p(X)}{p(\mathcal{D})}\right]$$

$$= \mathbb{E}_{q_\theta(X)}\left[\log q_\theta(X) - \log p(X) - \log p(\mathcal{D}|X)\right] + \log p(\mathcal{D})$$

$$= \mathbb{E}_{q_\theta(X)}\left[\log q_\theta(X) - \log p(X) - n\log C - \sum_{s=1}^{n}\log p(\{y_s, z_s\}|\{i_s, j_s, k_s\}, X)\right]$$
$$+ \log p(\mathcal{D})$$

$$= \mathbb{E}_{q_\theta(X)}\left[\log q_\theta(X) - \log p(X) - \sum_{s=1}^{n}\log p(\{y_s, z_s\}|\{i_s, j_s, k_s\}, X)\right]$$
$$- n\log C + \log p(\mathcal{D}).$$

### B.3 VICE objective function

Because $n\log C$ and $\log p(\mathcal{D})$ are constants and not functions of the variational parameters, we can ignore both terms in the minimization and get the following VICE objective

$$\arg\min_{\theta}\ \mathbb{E}_{q_\theta(X)}\left[\log q_\theta(X) - \log p(X) - \sum_{s=1}^{n}\log p(\{y_s, z_s\}|\{i_s, j_s, k_s\}, X)\right].$$

Multiplying this by $(1/n)$, where $n$ is the number of training examples, does not change the minimum of the objective function and results in

$$\arg\min_{\theta}\ \mathbb{E}_{q_\theta(X)}\left[\frac{1}{n}(\log q_\theta(X) - \log p(X)) - \frac{1}{n}\sum_{s=1}^{n}\log p(\{y_s, z_s\}|\{i_s, j_s, k_s\}, X)\right].$$

## C  Optimization, convergence and prior

### C.1 Gradient-based optimization

In gradient-based optimization, the gradient of an objective function with respect to the parameters of a model, $\nabla\mathcal{L}(\theta)$, is used to iteratively find parameters $\hat{\theta}$ that minimize that function. Equation 3 computes the expected log-likelihood of the entire training data. However, using every training data point to compute a gradient update is computationally expensive for large datasets and often generalizes poorly for non-convex objective functions [5]. In VICE, we stochastically approximate the training log-likelihood using random subsets (i.e., mini-batches) of the training data, where each mini-batch consists of $B$ triplets [4]. This leads to an objective function that is a doubly stochastic approximation of Equation 3 [6], due to sampling weights from the variational distribution, $q_\theta \in \mathcal{Q}$, and sampling a random mini-batch of training examples during each gradient step (i.e., performing mini-batch gradient descent),

$$\mathcal{L}_{batch} = \frac{1}{n}\left[\log q_\theta(X_{\theta,\epsilon}) - \log p(X_{\theta,\epsilon})\right] - \frac{1}{B}\sum_{b=1}^{B}\log p(\{y_b, z_b\}|\{i_b, j_b, k_b\}, [X_{\theta,\epsilon}]_+),$$

where $p(\{y, z\}|\{i, j, k\}, X)$ for a single sample is defined in Equation 1. To find parameters, $\theta$, that optimize Equation 3, we iteratively update both the means, $\mu$, and the standard deviations, $\sigma$, of each VICE dimension, by

$$\mu_{t+1} \coloneqq \mu_t - \alpha\nabla_{\mu_t}\mathcal{L}_{batch}$$

and

$$\sigma_{t+1} \coloneqq \sigma_t - \alpha\nabla_{\sigma_t}\mathcal{L}_{batch},$$

where $\alpha$ is the learning rate for $\theta$.

## C.2 Convergence

We define *convergence* as the point in time, $t^*$, where the number of embedding dimensions has not changed by a *single* dimension over the past $L$ epochs. We denote this as the point of *representational stability*. To ensure convergence, we recommend letting $L$ be relatively large (e.g., $L \gg 100$). We found $L = 500$ to work well for our experiments. Figure 1 shows that the convergence criterion defined in §5.2 worked reliably for all datasets.

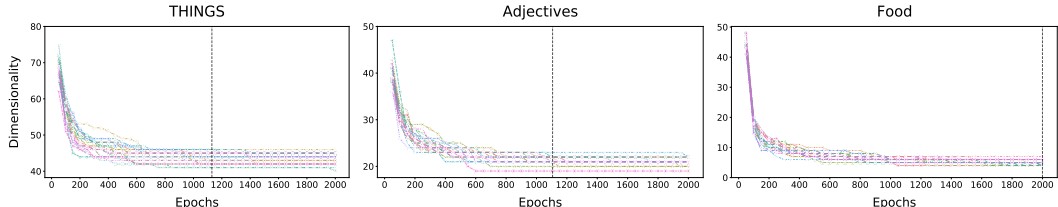

Figure 1: These plots show the number of embedding dimensions over time for THINGS, ADJECTIVES, and FOOD respectively. Each line in a plot corresponds to a single random seed. Vertical dashed lines indicate the median number of epochs (across random seeds) until the convergence criterion with $L = 500$ was met.

## C.3 (In-)efficient prior choice

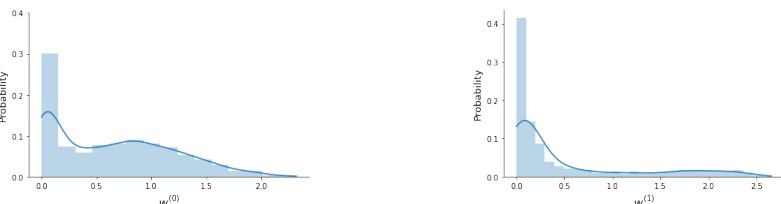

Figure 2: Histograms and PDFs of the first two SPoSE dimensions after training.

As discussed in §5.1, SPoSE imposes a combination of an $\ell_1$ penalty and a non-negativity constraint on its embedding values. This is analogous to having an exponential prior on those values. If we consider the distribution of values across objects for the two most important SPoSE dimensions in Figure 2, we can see that they show a distribution with a spike around $0$, and a much smaller, wide slab of probability mass for non-zero values. Overcoming the exponential prior is data inefficient. SPoSE was developed using a dataset that was orders of magnitude larger than what a typical psychological experiment might collect, a setting in which this issue would likely not manifest itself. However, SPoSE has not been tested on the more common, smaller datasets, and our results suggest that the implicit SPoSE prior leads to suboptimal results in those regimes, when compared to the spike-and-slab prior used by VICE, as shown in Figure 2.

## D  Proof of Proposition 3.1

*Proof.* Recall that

$$\hat{R}(X) := \sum_{s=1}^{n} \frac{1}{n} \mathbb{1}\left(\{y_s, z_s\} \neq \underset{\{y,z\}}{\arg\max}\, p\left(\{y, z\}|\{i_s, j_s, k_s\}, X\right)\right)$$

and

$$R(X) := P_{\{y',z'\}|\{i',j',k'\}}\left(\{y', z'\} \neq \underset{\{y,z\}}{\arg\max}\, p\left(\{y, z\}|\{i', j', k'\}, X\right)\right).$$

Using the union bound we have that

$$P\left(\sup_{X \in \mathbb{A}^{m \times d}} \hat{R}(X) - R(X) \geq \epsilon\right) \leq \sum_{X \in \mathbb{A}^{m \times d}} P\left(\hat{R}(X) - R(X) \geq \epsilon\right). \tag{2}$$

Since $\mathbb{E}\left[\hat{R}(X)\right] = R(X)$ and the summands of $\hat{R}(X)$ are independent and satisfy

$$0 \leq \frac{1}{n}\mathbb{1}\left(\{y_s, z_s\} \neq \underset{\{y,z\}}{\arg\max}\, p\left(\{y,z\}|\{i_s, j_s, k_s\}, X\right)\right) \leq \frac{1}{n}$$

for all $X \in \mathbb{A}^{m \times d}$ we can apply Hoeffding's Inequality to yield

$$P\left(\hat{R}(X) - R(X) \geq \epsilon\right) \leq \exp\left(-\frac{2\epsilon^2}{\sum_{i=1}^{n}\left(\frac{1}{n} - 0\right)^2}\right) = \exp\left(-2\epsilon^2 n\right) \qquad (3)$$

for all $X \in \mathbb{A}^{m \times d}$. The set $\mathbb{A}^{m \times d}$ contains $(k+1)^{md}$ elements so combining Equation 2 and Equation 3 we get

$$P\left(\sup_{X \in \mathbb{A}^{m \times d}} \hat{R}(X) - R(X) \geq \epsilon\right) \leq (k+1)^{md}\exp\left(-2\epsilon^2 n\right)$$

$$= \exp\left(md\log(k+1) - 2\epsilon^2 n\right)$$

$$\iff P\left(\sup_{X \in \mathbb{A}^{m \times d}} \hat{R}(X) - R(X) < \epsilon\right) \geq 1 - \exp\left(md\log(k+1) - 2\epsilon^2 n\right).$$

To arrive at the proposition statement we need that

$$1 - \delta \geq 1 - \exp\left(md\log(k+1) - 2\epsilon^2 n\right).$$

Solving for $n$ we get

$$1 - \exp\left(md\log(k+1) - 2\epsilon^2 n\right) \geq 1 - \delta$$

$$\iff \exp\left(md\log(k+1) - 2\epsilon^2 n\right) \leq \delta$$

$$\iff md\log(k+1) - 2\epsilon^2 n \leq \log(\delta)$$

$$\iff \left(md\log(k+1) + \log(1/\delta)\right)/(2\epsilon^2) \leq n.$$

$\square$

## E   Experimental details

**Training**   Although we have developed a reliable convergence criterion for VICE (see §5.2), to guarantee a fair comparison between VICE and SPoSE, each model configuration was trained, using 20 different random seeds for 2000 epochs. Each VICE model was initialized with two weight matrices, $\mu \in \mathbb{R}^{m \times d}$ and $\log(\sigma) \in \mathbb{R}^{m \times d}$, where $m$ refers to the number of unique objects in the dataset (THINGS: $m = 1854$; ADJECTIVES: $m = 2372$; FOOD: $m = 36$) and $d$, the initial dimensionality of the embedding, was set to 100 (the log ensures that $\sigma$ is positive). In preliminary experiments, we observed that for sufficiently large $d$ our dimensionality reduction method (see §5.2) prunes to similar representations, regardless of the choice of $d$. This is why we did not consider models with larger initial embedding dimensionality.

**Weight initialization**   We initialized the means of the variational distributions, $\mu$, following a Kaiming He initialization [1]. The logarithms of the scales of the variational distributions, $\log(\sigma)$, were initialized with $\epsilon = -1/s_\mu$, where $s_\mu$ is the standard deviation over the entires of $\mu$, so $\log(\sigma) = \epsilon\mathbf{1}$.

**Hyperparameter grid**   The final grid was the Cartesian product of the following hyperparameter sets: $\pi_{\text{spike}} = \{0.1, 0.2, 0.3, 0.4, 0.5, 0.6, 0.7, 0.8, 0.9\}$, $\sigma_{\text{spike}} = \{0.125, 0.25, 0.5, 1.0, 2.0\}$, $\sigma_{\text{slab}} = \{0.25, 0.5, 1.0, 2.0, 4.0, 8.0\}$, subject to the constraint $\sigma_{\text{spike}} \ll \sigma_{\text{slab}}$, where combinations that did not satisfy the constraint were discarded. We observed that setting $\sigma_{\text{slab}} > 8.0$ led to numerical overflow issues during optimization. For SPoSE, we used the same range as was done in Zheng et al. [7], with a finer grid of 64 values.

**Optimal hyperparameters**  We found the optimal VICE hyperparameter combination through a two step procedure. First, among the final $180$ combinations (see Cartesian product above), we applied our pruning method (see §5.2) to each model and kept the subsets of dimensions where more than 5 objects had non-zero weight. For SPoSE we used the pruning heuristic proposed in Zheng et al. [7]. We defined the optimal hyperparameter combination as that with the lowest average cross-entropy error on the validation set across twenty different random initializations. The optimal hyperparameter combinations for VICE and SPoSE on the full datasets are reported in Table 1.

Table 1: Optimal hyperparameter combinations for VICE and SPoSE according to the average cross-entropy error on the validation set for the three datasets THINGS, ADJECTIVES, and FOOD.

| DATA\ HYPERPARAM. | VICE | | | SPoSE |
| --- | --- | --- | --- | --- |
| | $\sigma_{\text{spike}}$ | $\sigma_{\text{slab}}$ | $\pi$ | $\lambda$ |
| THINGS | 0.125 | 0.5 | 0.6 | 5.75 |
| ADJECTIVES | 0.25 | 0.5 | 0.6 | 4.96 |
| FOOD | 0.25 | 1.0 | 0.8 | 2.90 |

**Computational resources**  This study utilized the high-performance computational capabilities of the Biowulf Linux cluster at the National Institutes of Health, Bethesda, MD (`http://biowulf.nih.gov`) and the Raven and Cobra Linux clusters at the Max Planck Computing & Data Facility (MPCDF), Garching, Germany (`https://www.mpcdf.mpg.de/services/supercomputing/`). The total number of CPU hours used were approximately $40,000,000$ (Biowulf) and $5,000$ (MPCDF).

**Accuracy upper bound**  For THINGS [7, 3] and ADJECTIVES, a random subset of triplets was chosen to be presented multiple times to different participants. For a given triplet - repeated over many participants - this provides a way to estimate the distribution of responses over all participants. If for a given triplet the response distribution is $(0.2, 0.3, 0.5)$, then the best predictor for the participants' responses is the third object. This results in an accuracy score of 50%, averaged across repetitions. Alternatively, one may observe a distribution of $(0.1, 0.8, 0.1)$ for a different triplet. The best one could do is to identify the second object as the odd-one-out, and get 80% accuracy. From this, we can see that no classifier can do worse than 33%. Taking the average best prediction accuracy over all of the repeated triplets gives us an estimate for the best possible average prediction score. This is defined to be the upper bound for the prediction performance.

# F   Generalization error bound

## F.1   Algorithm for generalization error upper bound

---
**Algorithm 1** Algorithm for generalization error upper bound via adaptive quantization

---
**Input:** $\mu$
  $M \leftarrow \max(\mu)$
  $\{\Delta_1, \ldots, \Delta_m\} \leftarrow \{0.05, ..., 1.0\}$                  ▷ pre-determined set of quantization scales
  $\alpha \leftarrow 0.05$                                           ▷ desired Type I error control rate
  $\delta \leftarrow \frac{\alpha}{m}$
  **for** $i \in \{1, \ldots, m\}$ **do**
    $\mu_i^\dagger := \text{quantize}(\mu, \Delta_i)$                          ▷ quantization with $\Delta_i$
    $\hat{R}_i = \text{loss}(\mu_i^\dagger, \mathcal{D})$                          ▷ (training) error
    $\bar{R}_i := \hat{R}_i + \sqrt{\frac{md \log(\lceil M/\Delta_i \rceil + 1) + \log(1/\delta)}{2 N_{\text{train}}}}$        ▷ generalization upper bound
  **end for**
  $i^* := \arg\min\{\bar{R}_i, ..., \bar{R}_m\}$
**Output:** $(\mu_{i^*}^\dagger, \bar{R}_{i^*})$                               ▷ $\bar{R}_{i^*}$ holds with probability $1 - \alpha$

---

We have the flexibility of choosing the quantization scale post-hoc. As long as we search over a pre-specified set of $m$ quantization scales $\{\Delta_1, \ldots, \Delta_m\}$, using a union bound, the PAC bound holds

simultaneously for all quantized embeddings with probability at least $1 - m\delta$. Therefore, we can find the quantization scale that gives us the best probabilistic upper bound on generalization error.

## F.2 Quantization

This section describes a number of empirical findings which support the feasibility of obtaining useful bounds by using our proposed quantization-based PAC bound (see §6) and the algorithm for obtaining retrospective generalization bounds (see Appendix F.1).

Recall that we require two assumptions for our bounding approach to be effective. Specifically, we assume,

1. Sparse embeddings obtained by either SPoSE or VICE can be quantized in a relatively coarse fashion, with only marginal losses in predictive performance.
2. There exists an upper bound $M$ on the largest value in an embedding.

In the following we present empirical results which demonstrate that those assumptions are satisfied for the three datasets, THINGS, ADJECTIVES, and FOOD, which we have used to evaluate VICE.

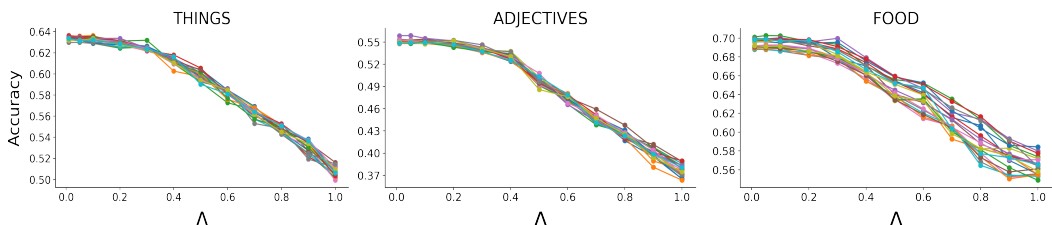

Figure 3: Test performance as a function of the quantization scale, $\Delta$, for twenty random initializations for THINGS, ADJECTIVES, and FOOD.

Recall that it is possible to choose a certain quantization scale, $\Delta$, and round an embedding value to a non-negative integer multiple of $\Delta$. While we employ quantization mainly to use STL bounds for finite hypothesis classes, it could have benefits for interpretation as well (e.g., a dimension would consist of labels such as zero (0), very low (0.5), low (1), medium (1.5), high (2)).

In Figure 3 we show generalization performances of VICE models as a function of the quantization scale, $\Delta$, for THINGS, ADJECTIVES, and FOOD. To quantize the VICE embeddings, we used the same set of quantization scales for every dataset, $\Delta \in \{0.05, \ldots, 1.0\}$, as defined in Algorithm 1. As $\Delta$ increases, the number of bins, to which the embedding values can possibly be assigned, decreases. That is, the space of possible embedding values gets smaller as a function of increasing $\Delta$, and, therefore, maintaining generalization performance becomes more difficult. Note that quantization with $\Delta = 0$ is equivalent to performing inference with the original embeddings.

As a result, the optimal generalization bounds that we get by using Algorithm 1 are typically obtained with $\Delta \le 0.2$. However, we explore a much larger range since *a priori* we do not know the level of granularity needed to preserve most of the information in an embedding. Theoretically, there exist embeddings, such as highly sparse embeddings with nearly binary elements, which could maintain high accuracy at large quantization scales. Hence, the optimal level of quantization for the bound is an empirical question that may vary across datasets. However, we limit the upper range to 1, because if the largest weight is less than 2.7 (as we will examine below), then $\Delta = 1$ already reduces the number of distinct weight values to 3, which is small enough that we are likely to see diminishing returns by considering even larger discretization scales.

**Upper bound** The upper bound, $M$, as defined in §6, for the VICE mean embeddings was approximately 2.7 for all datasets. Empirically, we found the maximum weights for THINGS, ADJECTIVES, and FOOD to be 2.6285, 2.2130 and 2.4307, respectively, across all random initializations. Theoretically, this upper bound is not surprising due to a combination of two factors. First, in most datasets, the cross-entropy error goes to infinity as weights become arbitrarily large, due to the increasing over-confidence of incorrect probability estimates. Hence, the optimal cross-entropy is achieved with bounded weights. Second, almost any kind of regularization, including the $\ell_1$ regularization

used in SPoSE and the spike-and-slab regularization used for VICE, further encourages the weights to shrink. The spike-and-slab prior, in particular, penalizes large weights much higher than small weights, which is a desirable property in gradient-based optimization.

# G  Dataset acquisition

All datasets were collected by crowd-sourcing human responses to the triplet task described in §3 on the Amazon Mechanical Turk platform, using workers located in the United States. For ADJECTIVES, all words deemed offensive were removed from the list of adjectives being considered. There are no offensive images in either THINGS or FOOD. All workers provided informed consent, and were compensated financially for their time ($0.5\ c$ per response, and additional $10\ c$ per completed HIT). Workers could participate in blocks of 20 triplet trials and could choose to work on as many such blocks as they liked. The online research was approved by the Office of Human Research Subject Protection at the National Institutes of Health and conducted in accordance with all relevant ethical regulations. Worker ages were not assessed, and no personally identifiable information was collected.

**THINGS and ADJECTIVES**    A total of 5,526 workers participated in collecting the first dataset (5,301 after exclusion; 3,159 female, 2,092 male, 19 other, 31 no response). A total of 336 workers participated in collecting the second dataset (325 after exclusion; 156 female, 103 male, 66 not reported). Workers were excluded if they exhibited overly fast responses in at least 5 sets of 20 trials (the speed cut-off was $25\%$ or more responses $< 800$ms and $50\%$ or more responses $< 1,100$ms) or if they carried out at least 200 trials and showed overly deterministic responses ($> 40\%$ of responses in one of the three odd-one-out positions; expected value, $33\%$).

**FOOD**    A total of 554 subjects participated in collecting the dataset (487 after exclusion). Workers were excluded if they exhibited overly fast responses (reaction time of less than 500ms). They were also excluded if they failed on either of two catch trials in each HIT, where they saw images of '+', '-', '-' or '=', '+', or '=' instead of food pictures; they were instructed on the slide to select the '+'.

# H  Interpretability

Here, we display the top 6 objects for each of the 45 VICE dimensions THINGS [2]. Objects were sorted in descending order according to their absolute embedding value. As we have done for every other experiment, we used the *pruned* median model to guarantee the extraction of a representative sample of embedding dimensions without being over-optimistic with respect to their interpretability (see §7.1 for how the median model was identified).

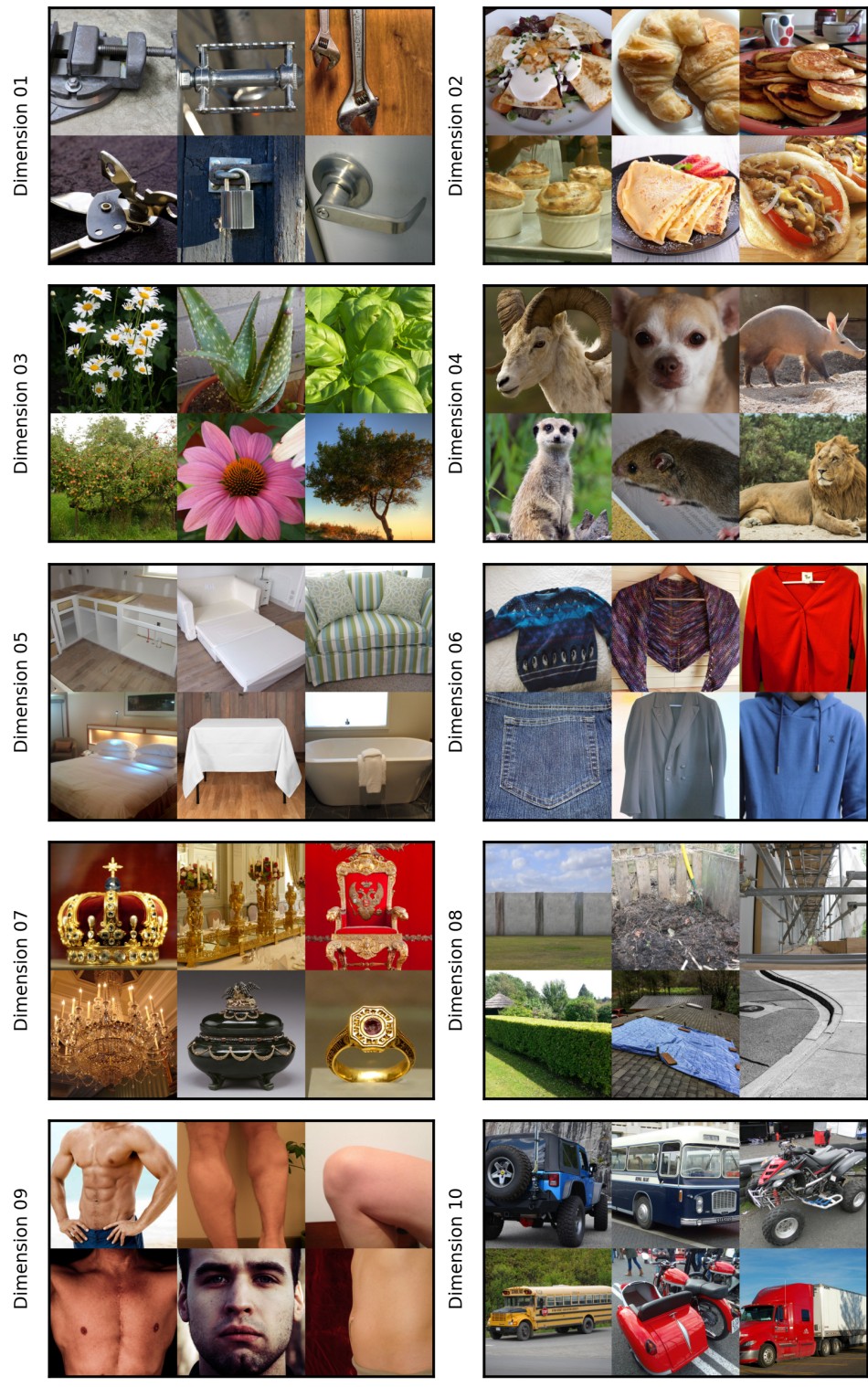

Figure 4: THINGS Dimensions 1-10.

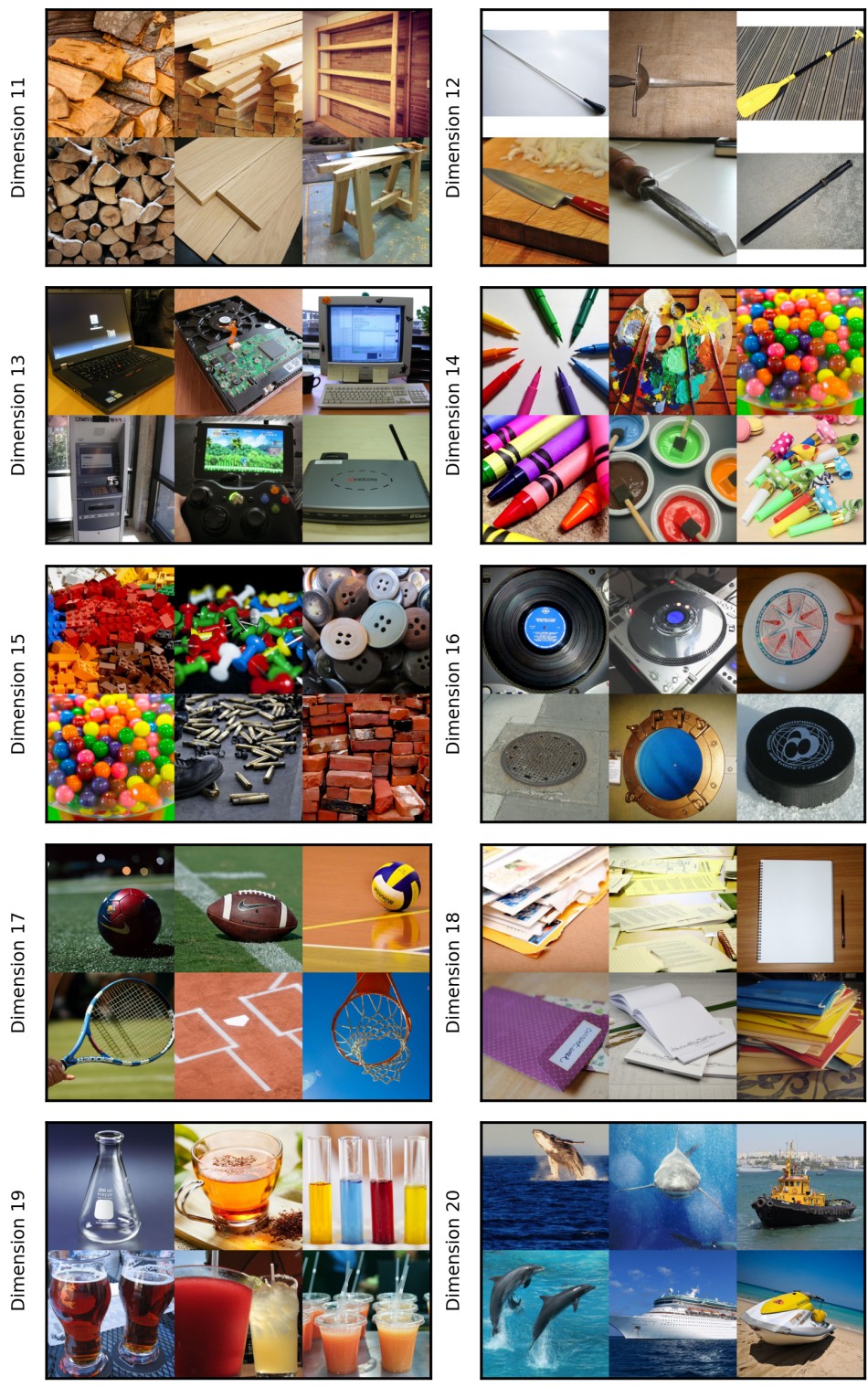

Figure 5: THINGS Dimensions 11-20.

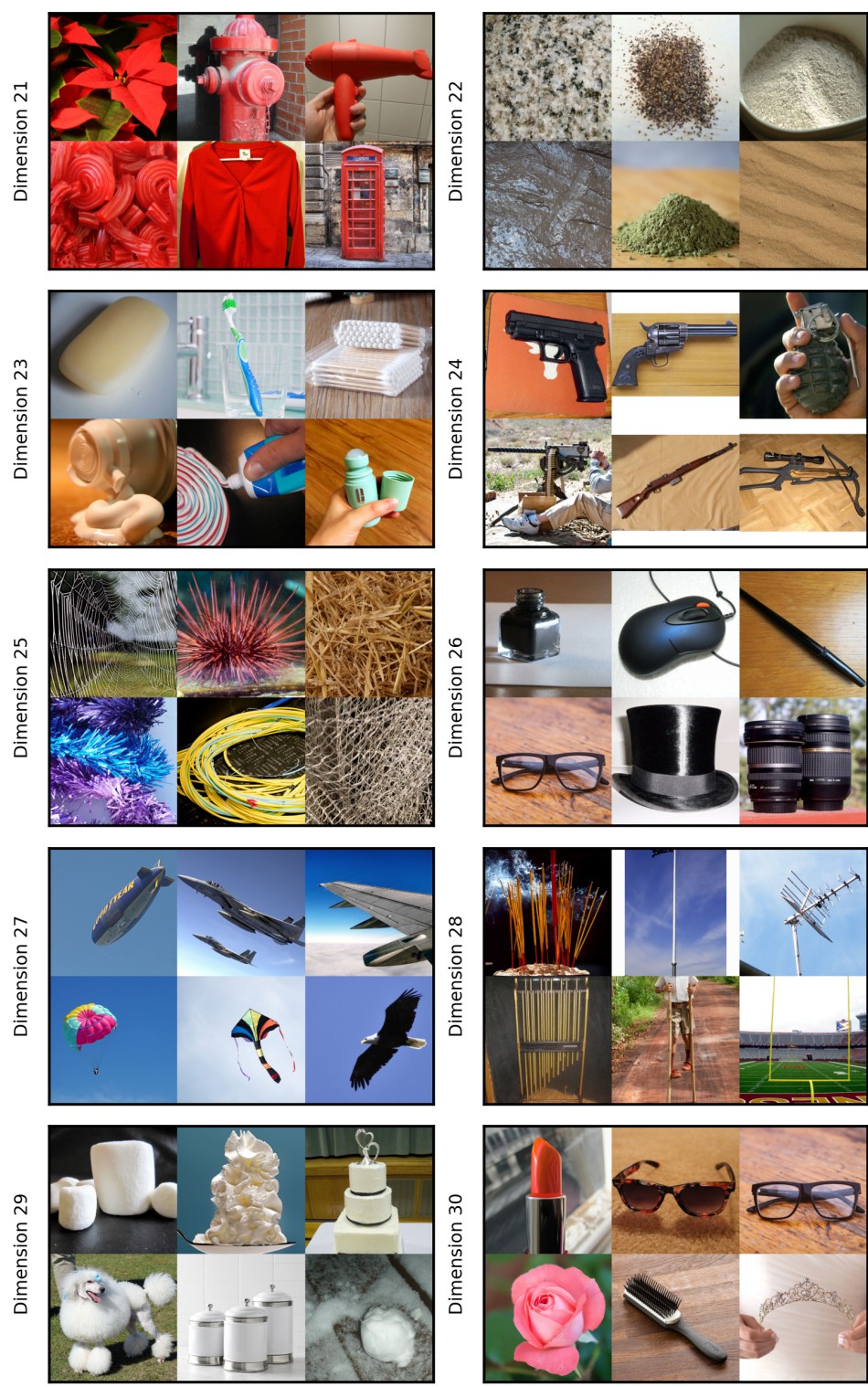

Figure 6: THINGS Dimensions 21-30.

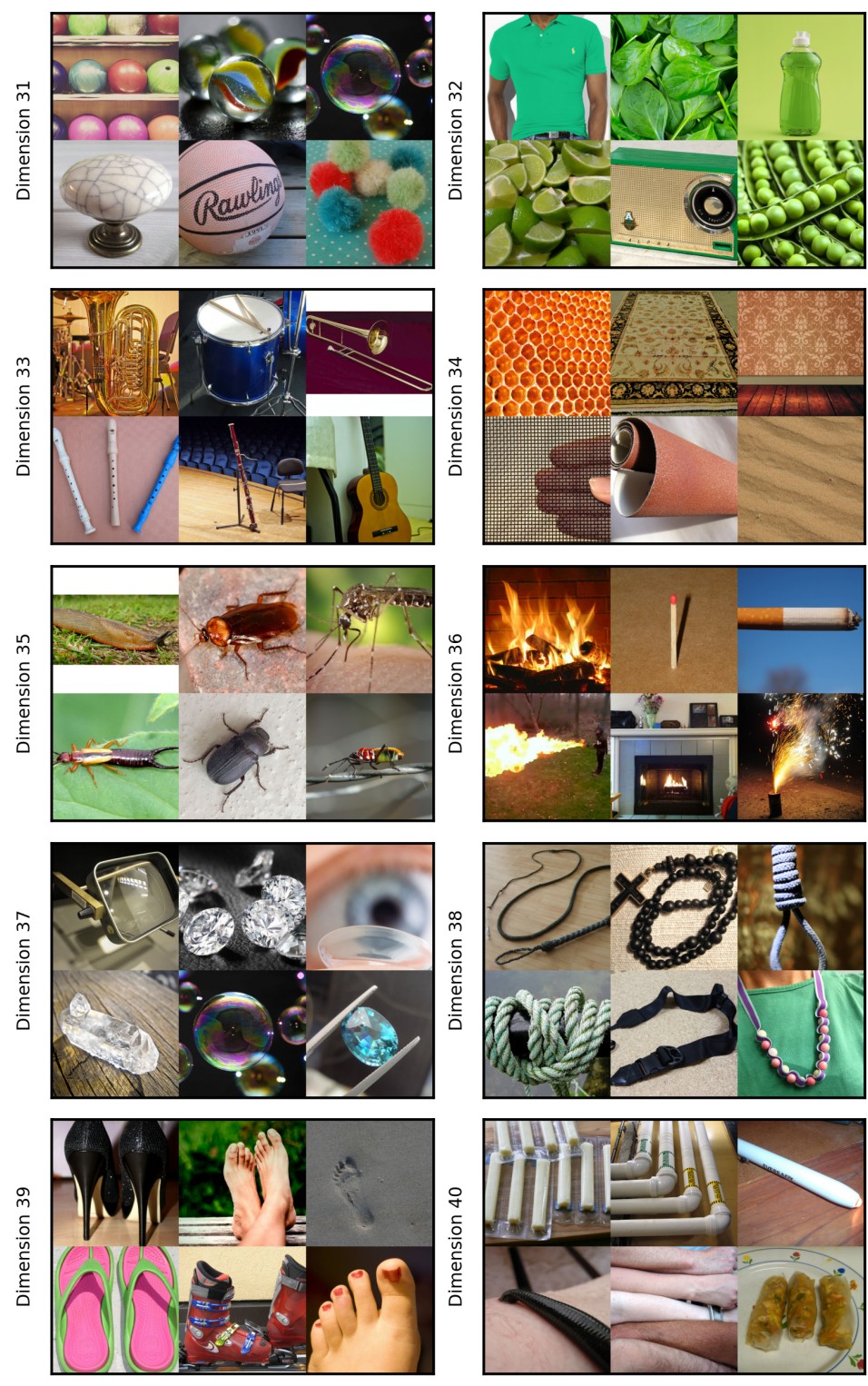

Figure 7: THINGS Dimensions 31-40.

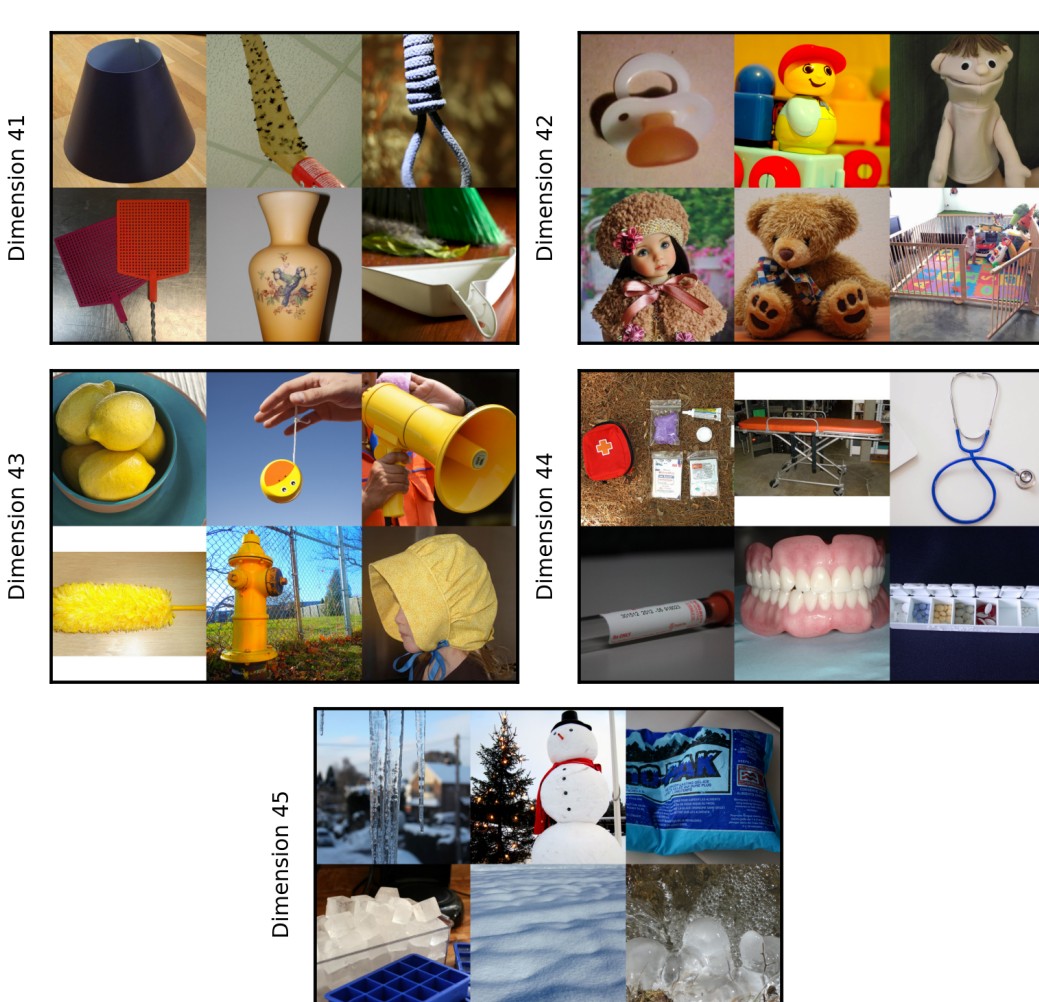

Figure 8: THINGS Dimensions 41-45.