# OpenReview forum: "VICE: Variational Interpretable Concept Embeddings"
_NeurIPS.cc/2022/Conference — NeurIPS 2022 Accept_

### Official Review · Reviewer_tLve · 2022-07-09

**Rating:** 6
**Confidence:** 3
**Soundness:** 4 excellent
**Presentation:** 3 good
**Contribution:** 3 good

**Summary:**

This work comprehensively analyzed the limitations of SPoSE from the aspects of the posterior distribution, estimation method, and convergence criterion. The proposed method VICE addressed the above limitations with elegant mathematical support. The authors reinterpreted the objective of SPoSE from the perspective of maximizing a posteriori (MAP) estimation, which is insightful for future work. The proofs are solid, especially dimensionality reduction and convergence. The experimental results were also positive to support the claims.

**Questions:**

- Since it is a supervised method, how about the efficiency of labeling a pair from a triplet? Why not label the concepts directly?
- Does the Spike-and-Slab prior for p(X) come from an empirical observation?
- Why the embeddings should be positive?

**Ethics Review Area:**

["I don’t know"]

**Limitations:**

This method needs numerous labels from human judgments. However, some work could promote dimensional explanation without supervision, such as disentanglement learning. The main limitation is the problem settings in the odd-one-out triplet task. Despite the limitations, I am looking forward to seeing the benefit of supervision and possible approaches to building large triplet datasets by automatic techniques.

Reference:
1. DEFT: Distilling Entangled Factors by Preventing Information Diffusion

**Strengths And Weaknesses:**

Strengths ：
+ This work explicitly described the introduced assumptions and how to induce the proposed method. This part is easy to read and understand.
+ This work originally proposed VICE with a unimodal posterior for representing each object and variational inference for optimization to learn from human judgments about object similarity.
+ This work encouraged interpretable representations, which is highly desirable for robust and trustworthy AI systems.

Weaknesses：
- It's unclear for the neural network settings and how to get the embedding vectors.
- What are the effects of the biases from judgments? It's important but missed. For instance, will the learned representations from ordinary people be different from those from specialists?

---

> ### Author Response · Authors · 2022-08-01
> **Author response**
>
> **It's unclear for the neural network settings and how to get the embedding vectors.**
>
> VICE has access to the human responses rather than to the image representations of the objects. From these responses, VICE learns an embedding representation for each object. Although there is a softmax function involved to get a probabilistic estimate for the odd-one-out choice in a triplet, given the learned embeddings, VICE does not use a neural network to map input to latent representations.
>
> Our method is related to Bayesian non-negative matrix factorization but uses variational inference and gradient descent instead of iterative coordinate descent solvers. VICE learns two weight matrices, one weight matrix for the means of the embedding values ($\mu$) and another weight matrix for the standard deviations of the embedding values ($\sigma$). Using the reparametrization trick (described in line 139), for each mini-batch, we sample an embedding matrix, $X \in \mathbb{R}^{m \times d}$, and select the object embeddings with one-hot vectors. Recall that each of the $m$ objects is assigned a numerical index. This is described in detail in §3.3.1.
>
> After convergence, one can use the means of the embedding values as representative object embeddings for interpretability purposes. This is what we did in §4.7.
>
> **Since it is a supervised method, how about the efficiency of labeling a pair from a triplet? Why not label the concepts directly?**
>
> **Some work could promote dimensional explanation without supervision, such as disentanglement learning. The main limitation is the problem settings in the odd-one-out triplet task.**
>
> We agree that it would be possible to use pre-existing methods to transform objects into embeddings (e.g., via a neural network), and that one could use disentangled representations for that purpose. This would be appropriate if the primary goal was to predict human behavior.  In our case, however, the odd-one-out task is meant to elicit the use of object-relevant knowledge without biasing the experimental participants. This is what makes it possible to identify embedding dimensions that explain behavior, without the need to postulate them *a priori*.
>
> **What are the effects of the biases from judgments? It's important but missed. For instance, will the learned representations from ordinary people be different from those from specialists?**
>
> As multiple reviewers raised this question, please see our general response.
>
> **Does the Spike-and-Slab prior for p(X) come from an empirical observation?**
>
> As multiple reviewers raised this question, please see our general response.
>
> **Why the embeddings should be positive?**
>
> In informal discussions with the authors of Zheng et al. (2019), we learned that the authors initially considered real-valued embeddings for objects. However, the result was finding a small number of very difficult-to-interpret embeddings, similar in properties to well-known word embeddings such as Word2Vec or synsets. Yet, when adding a non-negativity constraint, the number of dimensions was seen to increase, the prediction performance improved, and the interpretability of the dimensions was dramatically improved. This was not entirely surprising, given that the association between interpretability and non-negativity is widely observed in the literature across a large range of machine learning models. Intuitively, one can view a non-negative dimension as standing for the degree of presence of a particular characteristic. One theoretical account of this phenomenon is given in [4].
>
> With respect to concept embeddings, it is still not clear whether the success of non-negative embeddings is due to theoretical reasons owing to favorable mathematical properties of non-negative embeddings, or because the human brain actually uses non-negative representations.  In VICE embeddings on word stimuli, we have seen that properties that we might intuitively consider to lie on a positive-to-negative spectrum are sometimes represented in the model by two different, positive embeddings: one corresponding to the positive side of the spectrum, and the second corresponding to the negative.  While this is suggestive that the brain represents negative qualities as a separate scale from positive qualities, it is not possible to resolve these questions without further scientific investigation.
>
> **Reference**
>
> Last, but not least we thank the reviewer for pointing us to the interesting reference “DEFT: Distilling Entangled Factors by Preventing Information Diffusion.”
>
> **Final note**
>
> As a final note, we thank the reviewer for their feedback and for suggesting that we discuss the effects of potential biases in the data. We will discuss this further in the paper, as mentioned in the general response.
>
> **References**
>
> [4] Donoho, David, and Stodden, Victoria. "When does non-negative matrix factorization give a correct decomposition into parts?" Advances in neural information processing systems 16 (2003).

---

### Official Review · Reviewer_fyj5 · 2022-07-11

**Rating:** 5
**Confidence:** 3
**Soundness:** 2 fair
**Presentation:** 3 good
**Contribution:** 3 good

**Summary:**

This paper introduces VICE, a variational inference method for embedding object concepts into a vector space so as to obtain sparse and non-negative representations of them. VICE follows the same approach as its predecessor (SPoSE) to model the odd-one-out triplet task, but addresses SPoSE's major limitations by taking advantage of recent VI techniques. In addition, it also derives a PAC learning bound for SPoSE and VICE models, which can be used to estimate generalization performance or determine sufficient sample size.

**Questions:**

(1) The main difference from SPoSE need to be highlighted, technically.


**Strengths And Weaknesses:**

Pros:

The paper is overall clearly written and well-organized. The experiment section clearly states the setup/evaluation protocol and presents detailed analyses.

Cons:

Overall, I think the novelty of this paper is limited. Essentially, the idea of learning a sparse, positive (and hence interpretable) semantic space to be consistent with human similarity judgements (or finish the odd-one-out triplet task) is no different from SPoSE (by Charles et al). Following the same modeling process, VICE just makes some improvements over SPoSE with the help of VI techniques. The choice of Gaussian variational distributions for ease of reparameterization and the use of a spike-and-slab prior to induce sparsity are also based on previous successful experiences, so the ideas are not new there. It seems that the real novelty comes more from the derivation of a PAC bound on the generalization of SPoSE and VICE models, but I’m not sure I understand it since the author does not present the proof clearly enough.

---

> ### Author Response · Authors · 2022-08-01
> **Author response**
>
> **Overall, I think the novelty of this paper is limited. Essentially, the idea of learning a sparse, positive (and hence interpretable) semantic space to be consistent with human similarity judgements (or finish the odd-one-out triplet task) is no different from SPoSE (by Charles et al). Following the same modeling process, VICE just makes some improvements over SPoSE with the help of VI techniques. The choice of Gaussian variational distributions for ease of reparameterization and the use of a spike-and-slab prior to induce sparsity are also based on previous successful experiences, so the ideas are not new there.**
>
> The novel inference method is the primary contribution of this paper. We have demonstrated that it meets the state-of-the-art for prediction performance on current datasets while producing more stable representations with lower dimensionality. Furthermore, our method substantially outperforms previous methods in low data regimes. This is of great interest to the cognitive science community since data collection can be very expensive and thus our method is likely to have a high impact.
>
> Moreover, Bayesian extensions of effective models are of general interest to the ML community.  See for example the following papers that offer Bayesian approaches to matrix/tensor methods ($S \coloneqq XX^{T}$ in our paper in line 109 is low-rank, so our method is related):
>
> - Modelling Relational Data using Bayesian Clustered Tensor Factorization (NeurIPS 2009)
> - Statistical mechanics of low-rank tensor decomposition (NeurIPS 2018)
> - A Particle-Based Variational Approach to Bayesian Non-negative Matrix Factorization (JMLR 2019)
>
> The main (technical) differences between SPoSE and VICE are described in §1, lines 40-63, but we will highlight them further in the main text if our paper gets accepted.
>
> **It seems that the real novelty comes more from the derivation of a PAC bound on the generalization of SPoSE and VICE models, but I’m not sure I understand it since the author does not present the proof clearly enough.**
>
> The PAC bound for this type of model is indeed novel. We have added a detailed proof of the proposition to the appendix, which we reference in the main text. With the full proof, the bound is actually slightly improved, which we have also incorporated into the main text and into the quantization algorithm in the Appendix.
>
> **Final note**
>
> As a final note, we thank the reviewer for their feedback and for suggesting to clarify the proof of the PAC bound on the generalization performance of SPoSE and VICE models. Through their suggestion, we were able to improve the bound.

---

### Official Review · Reviewer_NZMy · 2022-07-12

**Rating:** 5
**Confidence:** 3
**Soundness:** 3 good
**Presentation:** 3 good
**Contribution:** 2 fair

**Summary:**

In this paper, a model named VICE is introduced for learning interpretable object concept embeddings by modelling human behaviour in an odd-one-out triplet task. It is shown that VICE predicts human behaviour close to the estimated best attainable performance across three datasets and that VICE outperforms a competing method SPoSE, in low sample regimes. In addition, the authors claim that VICE has an automated procedure for determining the number of dimensions needed to explain the data.

**Questions:**

Please refer to the previous section for detailed comments and discussion.

**Limitations:**

yes

**Strengths And Weaknesses:**

Strength:
1. The paper is relatively well written and the proposed method is easy to follow and understand.
2. The proposed method VICE is technically correct, including modelling definition, equations and derivations.
3. Both qualitative and quantitative comparisons are given to showcase the modelling performance and certain properties of the proposed method.
4. There is a lot of discussion on the intuition of the design of VICE, and discussion on comparison between VICE and SPoSe. I really enjoyed reading these discussions.
Weakness:
1. The paper claims to work on a very novel task of modelling human behaviour in an odd-one-out triplet task. However, after reading the description, it seems like a common classification task. You are given three input, and a set of two as label. It would be informative if the authors can expand the discussion on the task description part.
2. Related to point 1, it is also not clear how this task is associated with the concept embedding goal being evaluated. It is completely ok if learning a interpretable concept embedding is a nice side benefit of the task, or vice versa, but it would be clearer if the authors discuss this.
3. The paper needs more proof reading. For example, line 22 "In an alternative, detective, approach, ..." it look me a while to figure out the grammar here and to understand this sentence.
4. Triplet task seems to be a classification task. In line 104, the authors mentioned that the dataset is a set of ordered pairs. Why does the pairs need to be ordered? Does the ordering encode certain information?
5. Prior choice. Spike-and-Slab is chosen in this work based on analysis and intuition. Since this is a major novelty component of the proposed model, it would be more convincing if the authors could provide some experiments on the choice of prior. Such a comparison would back up the analysis on previous work and support the choice of Spike-and-Slab as prior.
6. The authors claim that one of the property of VICE is that VICE has an automated procedure for determining the number of dimensions needed to explain the data. Therefore, there isn't much need for running lots of configurations to tune hyper parameter. However, reading section 3.3.2, there is still a large number of hyper parameters. Even if there is a commonly used threshold present, it doesn't erase the fact that these hyper parameters still needs to be tuned to achieve the best modelling performance.

---

> ### Author Response · Authors · 2022-08-01
> **Author response 1/2**
>
> **"The paper claims to work on a very novel task of modelling human behaviour in an odd-one-out triplet task... it is also not clear how this task is associated with the concept embedding goal being evaluated. It is completely ok if learning a interpretable concept embedding is a nice side benefit of the task, or vice versa, but it would be clearer if the authors discuss this."**
>
> The problem that we address in the paper is understanding mental representations of objects, through the embedding vectors discovered through our model. These object embeddings are of wide scientific interest, as demonstrated by the citations of the Nature Human Behavior [1] article introducing SPoSE (a journal extension of [Zheng et al 2019]). Obtaining interpretable embeddings is the primary motivation of our work, not a side benefit. Indeed, we only consider model performance as a secondary consideration compared to the embeddings themselves: the performance of the model is informative to us only to the extent that it validates our assumptions about the data (e.g. spike and slab prior, sparsity), and demonstrates efficient use of the human behavior data  As such the novelty of our approach lies primarily in our principled use of variational techniques and technical innovations, including the development of pruning procedures and convergence criteria, that optimize the reproducibility of the embeddings, rather than our adherence to any particular behavioral task.
>
> We model the triplet task because cognitive scientists deem it to be a good way of probing a participant’s intuition about object similarities, without biasing them. An alternative approach would be to ask participants for numerical ratings of similarity (e.g. on a scale of 0-10) for pairs of objects, instead of triplets. However, participants may differ in how they calibrate their rating scales and are often inconsistent. Two participants with potentially the same notion of object similarity might give quite different numerical ratings, because each person's way of converting their intuitive sense of similarity to a numerical rating may be accomplished through an individually unique and somewhat arbitrary mapping.  In contrast, two participants with the same internal representations of object similarities should be able to agree with regard to which object is the least similar to the other two.
>
> **“The paper needs more proof reading. For example, line 22 "In an alternative, detective, approach, ..." it look me a while to figure out the grammar here and to understand this sentence.”**
>
> Thanks for pointing out the confusing comma usage. We’ve cleaned up our comma usage in line 22 and elsewhere.
>
> **“Triplet task seems to be a classification task. In line 104, the authors mentioned that the dataset is a set of ordered pairs. Why does the pairs need to be ordered? Does the ordering encode certain information?”**
>
> For a sample in the training data $\mathcal{D}$, the first entry contains the objects presented to the participant and the second entry contains the two objects selected as being the most similar. So, there technically is an ordering. We’d be happy to remove “ordered” if you feel it is less likely to be confusing.
>
> **Prior choice. Spike-and-Slab is chosen in this work based on analysis and intuition. Since this is a major novelty component of the proposed model, it would be more convincing if the authors could provide some experiments on the choice of prior. Such a comparison would back up the analysis on previous work and support the choice of Spike-and-Slab as prior.**
>
> As this concern was also raised by another reviewer, please see our general response.

---

> > ### Author Response · Authors · 2022-08-01
> > **Author response 2/2**
> >
> > **The authors claim that one of the property of VICE is that VICE has an automated procedure for determining the number of dimensions needed to explain the data. Therefore, there isn't much need for running lots of configurations to tune hyper parameter. However, reading section 3.3.2, there is still a large number of hyper parameters. Even if there is a commonly used threshold present, it doesn't erase the fact that these hyper parameters still needs to be tuned to achieve the best modelling performance.**
> >
> > At this point, we would like to reiterate that the _learned representations_ are the central object of interest in this work. These representations are of significant interest to cognitive scientists [1]. One wants good modeling performance on the classification task since this indicates that they have captured an understanding of how humans relate objects. If one were to find a classifier that has the absolute best classification performance, then we would agree with your suggestion, that one should consider using many dimensions, even hundreds. However, this would be of little use for cognitive scientists, since there would be a great loss of interpretability.
> >
> > The cut-off value $k$ is not a data-dependent hyperparameter, it is a standard threshold [2,3] for the minimum number of items that compose a psychologically relevant dimension. We find that the recovered representations are insensitive with respect to pi, and the scales for the spike and slab distributions, and so is predictive performance (see §4.5). For sufficiently large d we observed that the method consistently prunes to similar representations, regardless of the choice of d (note the low variance in Table 1 “Selected Dims.”). This point regarding sufficiently large d is mentioned in Appendix D, but not in the main text. In light of your comment, we’ve decided to include mention of this in the main text.
> >
> > **Final Note**
> >
> > We thank the reviewer for their feedback, and respectfully ask if they might consider increasing their score in light of our responses.
> >
> > **References**
> >
> > [1] Hebart, Martin N., et al. "Revealing the multidimensional mental representations of natural objects underlying human similarity judgements." Nature human behaviour 4.11 (2020): 1173-1185.
> >
> > [2] Barry J. Devereux, Lorraine K. Tyler, Jeroen Geertzen, and Billi Randall. The centre for speech, language and the brain (CSLB) concept property norms. Behavior Research Methods, 46 (4):1119–1127, December 2013. doi: 10.3758/s13428-013-0420-4.
> >
> > [3] Ken McRae, George S. Cree, Mark S. Seidenberg, and Chris Mcnorgan. Semantic feature production norms for a large set of living and nonliving things. Behavior Research Methods, 37 (4):547–559, November 2005. doi: 10.3758/bf03192726.438

---

### Official Review · Reviewer_VDhE · 2022-07-16

**Rating:** 6
**Confidence:** 1
**Soundness:** 3 good
**Presentation:** 3 good
**Contribution:** 3 good

**Summary:**

The paper proposed VICE, a way to learn interpretable embeddings that mimic that of a human. The paper is different from its main baseline (SPoSE) by using variational inference with a "spike and slab" Gaussian prior. The paper improves upon SPoSE by not having a hyperparameter that finds the # of embedding dimensions to ensure sparsity. VICE is able to improve upon SPoSE in the low-data regime, which the paper argues, is the most relevant setting in cognitive science.

**Questions:**

Please see above.

**Limitations:**

The authors do not discuss any societal limitations. It would be good to discuss if in the compression process, if it is possible that the model is implicitly encoding any biases that may exist in the data, since its directly labeled by humans, and there is not an absolute "ground truth" available for such datasets.

**Strengths And Weaknesses:**

Strengths:
- The paper seems theoretically sound as it improves considerably over the baselines
- The paper is written and explained well and has a clear story
- The results in the low data regime are very convincing as the model improves performance by ~20% in the lowest-data regime.\
- The dimensionality reduction procedure and spike-and-slam priors used by the paper is interesting.

Weaknesses:
- It would be preferred if the paper expanded on Fig 1, to give readers a better intuitive example of what the structure of the dataset is and some gt labels in the dataset, since it isn't obvious which one of the thruple is the "odd-one-out"
- Qualitative results for both SPoSE and VICE would be helpful (instead of just for VICE
- I am not sure if there are other baselines possible for such a paper (such as sysnet and NNSE used in Zhang et al.)
- I am not sure what the authors mean by: "We estimate an upper bound on the prediction accuracy by using the repeats in the test set." How is the upper bound in accuracy found?

---

> ### Author Response · Authors · 2022-08-01
> **Author response**
>
> **Qualitative results for both SPoSE and VICE would be helpful**
>
> In addition to the VICE dimensions, we added qualitative results for SPoSE to the supplementary material of our revised manuscript.
>
> **I am not sure if there are other baselines possible for such a paper (such as sysnet and NNSE used in Zhang et al.)**
>
> [Zheng et al., 2019] compared the performance of SPoSE vectors with synset vectors and NNSE vectors in terms of accuracy at predicting behavior in the odd-one-out-task and other tasks. These are reasonable baselines, in that synset vectors are the only text-derived embeddings that pertain to each object (rather than the word naming it), and NNSE vectors are a text-derived embedding that is sparse and positive. However, as both synset vectors and NNSE vectors generally performed worse than SPoSE in [Zheng et al.],  we decided against using them as a baseline in this paper, since the focus is on comparing SPoSE and VICE. If the paper is accepted, we will add a note to this effect in the discussion section of the paper.
>
> **I am not sure what the authors mean by: "We estimate an upper bound on the prediction accuracy by using the repeats in the test set." How is the upper bound in accuracy found?**
>
> For the human response data for THINGS (Zheng et al. 2019; Hebart et al., 2020) and Adjectives, a random subset of triplets was chosen to be presented multiple times to different participants. For a given triplet - repeated over many participants - this provides a way to estimate the distribution of responses over all participants. If the response distribution is (0.2, 0.3, 0.5) for a given triplet, then the best predictor for the participants' responses is the third object. This results in an accuracy score of 50%, averaged across repetitions. Alternatively, one may observe a distribution of (0.1,0.8,0.1) for a different triplet. The best one could do is to identify the second object as the odd-one-out, and get 80% accuracy. From this, we can see that no classifier can do worse than 33%.. Taking the average best prediction accuracy over all of the repeated triplets gives us an estimate for the best possible average prediction score. This is the upper bound.
>
> We’ve added this to the supplementary material for clarity.
>
> **The authors do not discuss any societal limitations. It would be good to discuss if in the compression process if it is possible that the model is implicitly encoding any biases…**
>
> Multiple reviewers raised this question, please see our general response.
>
> **Final note**
>
> We thank the reviewer for their feedback and their suggestion on discussing potential societal limitations. As mentioned in the general response, we will add further discussion of this to the paper.

---

### Author Response · Authors · 2022-08-01
**General response**

We would like to thank all reviewers for taking the time to provide thoughtful reviews. We were pleased to see that the reviewers unanimously recommended the paper for acceptance and found many positive aspects to our paper.

**VDhE**: “The paper seems theoretically sound as it improves considerably over the baselines. It is written and explained well and has a clear story.”

**NZMy**: “There is a lot of discussion on the intuition of the design of VICE, and discussion on comparison between VICE and SPoSe. I really enjoyed reading these discussions.”

**fyj5**: “The paper is overall clearly written and well-organized. The experiment section clearly states the setup/evaluation protocol and presents detailed analyses.”

**tLve**: “This work encouraged interpretable representations, which is highly desirable for robust and trustworthy AI systems.”



Here we address a couple of points that were shared among more than one reviewer. We have also provided individual reviewer responses.

**-Potential biases in the data-**

**Reviewer VDhE**: “The authors do not discuss any societal limitations… it is possible that the model is implicitly encoding any biases that may exist in the data, since its directly labeled by humans, and there is not an absolute "ground truth"

**Reviewer tLve**: “What are the effects of the biases from judgments? It's important but missed. For instance, will the learned representations from ordinary people be different from those from specialists?”

**Response**: The goal of our method is to identify general mental representations of objects. The dimensions identified by a model reflect semantic characteristics that explain task performance for many subjects in the experimental subject population (Amazon Mechanical Turk subjects in the United States). As such, it is possible that they reflect biases widespread in that population. Furthermore, the choice of population may affect the identified dimensions. That is, a chéf may classify food items differently from a lay subject, and a linguist would likely have a more complex representation of an adjective. The effects of expertise or developmental stage in mental representations are of obvious interest to cognitive scientists. Therefore, we envision further research in those areas, which may additionally provide some indication of how widely representations can vary.

We will add an additional paragraph including these points to a revised version of the manuscript.

**-The use of a spike-and-slab prior-**

**Reviewer NZMy**: “Prior choice. Spike-and-Slab is chosen in this work based on analysis and intuition. Since this is a major novelty component of the proposed model, it would be more convincing if the authors could provide some experiments on the choice of prior. Such a comparison would back up the analysis on previous work and support the choice of Spike-and-Slab as prior.”

**Reviewer tLVE**: “Does the Spike-and-Slab prior for p(X) come from an empirical observation?”

**Response**: Firstly, we have tried a Laplace prior and our results on the validation data were not better than SPoSE. This was our original motivation to consider other priors. Secondly, spike-and-slab priors are commonly used in the literature as a sparsity-inducing prior alternative to the Laplace prior. Thirdly, the distribution of SPoSE dimensions empirically appeared to be better modeled by a spike-and-slab distribution. This is described in §3.3.1, lines 149-154,


“As discussed above, SPoSE induces sparsity through an $\ell_{1}$ penalty which, along with the non-negativity constraint, is equivalent to using an exponential prior. Through examination of the publicly available histograms of weight values in the two most important SPoSE dimensions (see Figure 2 in B.3), we observed that the dimensions did not resemble an exponential distribution. Instead, they contained a spike of probability at zero and a wide slab of probability for the non-zero values. To model this, we use a *spike-and-slab* Gaussian mixture prior.”

Lastly, if a zero-mean Gaussian prior was better suited to our data than the spike-and-slab Gaussian mixture prior, we should have seen the components of the mixture collapse into one of the two Gaussians, which we did not observe.

---

### Meta-Review · Area_Chair_CCX8 · 2022-08-31

**Recommendation:** Accept
**Confidence:** Certain

**Metareview:**

This paper proposes a method to learn meaningful representations of data by incorporating a pick-odd-one-out task on triplets of images to learn embeddings through variational inference  using a spike-and-slab Gaussian prior.

The reviewers agreed that the paper was well written, had a clear narrative, that the results appeared convincing and that the use of the spike-and-slab prior to determine appropriate dimensionality was novel and interesting.

Where pertinent questions were raised by reviewers on the exposition of the estimation of the upper bound, on the validity of the triplet task and its phrasing, and on the utlity of employing VI over a prior model (SPoSE), the authors provided responses that appear to address these issues reasonably.

The primary issue with the manuscript appears to be mainly with framing. A graphical model, with annotations for the triplet observations, or something similar---a figure to explain the model basically---would have helped make things a bit easier to situate for the reader.

On balance, though it appears the paper has more merits than issues. I would strongly urge the authors to actually make the edits discussing other baselines (Reviewer VDhE), and potentially having the discussion on determining the number of latent dimensions in the main paper rather than the supplement as it appears to be an important distinguishing feature over prior work.


**Award:**

No

---

### Decision · Program_Chairs · 2022-09-14

Accept